# Retaining Beneficial Information from Detrimental Data for Deep Neural Network Repair

**Long-Kai Huang[1], Peilin Zhao[1], Junzhou Huang[2], Sinno Jialin Pan[3]**
[1]Tencent AI Lab
[2]The University of Texas at Arlington
[3]The Chinese University of Hong Kong
hlongkai@gmail.com, masonzhao@tencent.com, jzhuang@uta.edu, sinnopan@cuhk.edu.hk

## Abstract

The performance of deep learning models heavily relies on the quality of the training data. Inadequacies in the training data, such as corrupt input or noisy labels, can lead to the failure of model generalization. Recent studies propose repairing the model by identifying the training samples that contribute to the failure and removing their influence from the model. However, it is important to note that the identified data may contain both beneficial and detrimental information. Simply erasing the information of the identified data from the model can have a negative impact on its performance, especially when accurate data is mistakenly identified as detrimental and removed. To overcome this challenge, we propose a novel approach that leverages the knowledge obtained from a retained clean set. Concretely, Our method first identifies harmful data by utilizing the clean set, then separates the beneficial and detrimental information within the identified data. Finally, we utilize the extracted beneficial information to enhance the model's performance. Through empirical evaluations, we demonstrate that our method outperforms baseline approaches in both identifying harmful data and rectifying model failures. Particularly in scenarios where identification is challenging and a significant amount of benign data is involved, our method improves performance while the baselines deteriorate due to the erroneous removal of beneficial information.

## 1 Introduction

The effectiveness of deep neural networks (DNNs) is greatly influenced by the quantity and quality of the training data from which they are trained. However, obtaining a large amount of high-quality data can be costly. An alternative approach is to acquire training data from sources such as the Web [2], external data suppliers, or crowdsourcing [14]. Although this allows for the collection of a substantial volume of data, the quality of the acquired data cannot be guaranteed. Various issues can arise, including annotation errors [3], data corruption [13], or even deliberate adversarial manipulation [39] by malicious providers. Consequently, a portion of the collected data may have a different distribution than the majority of the data [21]. In this work, we refer to these two categories of data as "detrimental" and "target" data, respectively. The presence of detrimental data in the training set can significantly hinder the performance of a DNN [1]. While recent research has proposed methods to mitigate specific issues like label noise [31, 5, 11], adversarial attacks [28, 39], or data shifts during model training [21, 32], it remains challenging to train a model that generalizes well to target data when the specific cause of the detrimental data is unknown.

It is important to note that in practical scenarios, a DNN model must undergo a quality assurance test before it can be deployed. During this evaluation process, the performance of the trained DNN model is usually assessed using a small reserved set of data known as clean data. This clean data is expected

to have the same distribution as the target data. However, since the training process involves the use of detrimental data, the trained DNN model may produce inaccurate predictions on the clean data, leading to a failure in the quality assurance test. In such cases, it becomes necessary to repair the DNN model to successfully pass the quality assurance test.

Recent approaches to model repair focus on two main steps. Firstly, they aim to identify the detrimental data instances within the original training dataset with a set of clean data, a.k.a, test data. Once identified, these instances can be categorized for further action. Secondly, the model is updated by eliminating the identified detrimental data instances. This repair process helps in improving the model's performance and ensuring that it meets the desired quality standards for deployment. Notable works in this area include studies by Tanno et al.[33] and Kong et al.[23].

In contrast to previous approaches that focus on eliminating the entire detrimental dataset to learn a repaired DNN model, our approach is motivated by the belief that the detrimental data may contain some valuable underlying information. Once this information can be extracted, it can be leveraged to train a more generalized model alongside the target data. Drawing inspiration from domain adaptation techniques [35, 30], we propose a framework that aligns the identified detrimental data with the clean data. By aligning these datasets, we can utilize the aligned detrimental data to repair the DNN model and enhance its generalization ability, all without the need for complete retraining.

To elaborate, our proposed framework begins with an approach rooted in generalization theory to identify a potential set of detrimental data with the assistance of a clean data set, while simultaneously updating the model by mitigating the influence of the detrimental data during training. Subsequently, we introduce an energy-based algorithm [18] to align the identified detrimental data with the clean data. Through this alignment process, we selectively filter out harmful information that could adversely impact the model's generalization performance, while retaining the beneficial information that can bolster the model's overall generalization capabilities. By adopting this approach, we aim to strike a balance between effectively utilizing the beneficial aspects of the detrimental data and minimizing its negative impact on the model's performance. Our proposed framework offers a lightweight alternative to retraining the model from scratch, enabling the model to learn from both the target and aligned detrimental data, thereby improving its ability to generalize effectively.

It is crucial to acknowledge that identifying a perfect detrimental dataset from the training data is an arduous task, if not an impossible one. Previous approaches may discard useful information if certain target data instances are mistakenly identified as detrimental. In our proposed framework, we mitigate this issue by employing an energy-based alignment algorithm to reduce the divergence between the aligned detrimental data and target distributions. In essence, the alignment process serves as a corrective mechanism for potential errors introduced during the detrimental data identification step. By aligning the identified detrimental dataset with the clean data, our framework can rectify misidentifications and ensure that the model benefits from the valuable information contained within the misidentified instances. By leveraging the alignment process, our framework achieves a higher level of resilience, ensuring that the model's generalization performance remains intact, even in the presence of potential misidentifications. This robustness sets our proposed approach apart from previous methods and reinforces its effectiveness in dealing with the challenges associated with identifying and utilizing detrimental data.

**Summary of contribution**: We introduce a novel framework that acknowledges the potential value of detrimental data in enhancing the training of a more generalized model alongside target data. Our framework aligns the identified detrimental data with clean data, enabling us to leverage this aligned data to repair the DNN model and improve its generalization ability without the need for complete retraining. This approach strikes a balance between utilizing the beneficial aspects of detrimental data and minimizing its negative impact. By rectifying potential misidentifications through alignment, our framework maximizes the utilization of both target and aligned detrimental data, resulting in a more robust and resilient approach compared to previous methods for addressing the challenges associated with utilizing detrimental data. We conducted extensive experiments on real-world datasets to validate the effectiveness of our proposed framework, demonstrating its superiority over existing methods in identifying harmful data and repairing the model to enhance generalization.

## 2 Methodology

**Problem Settings**: Assume our goal is to maximize model performance on some target data distribution $\mathcal{Z}$ and we have trained a prediction model $\hat{\theta}$ by minimizing the empirical risk $\mathcal{L}(\mathcal{D}; \theta)$ over a training set $\mathcal{D} = \{z_i\}_{i=1}^{N}$ consisting of $N$ data pairs $z_i = (x_i, y_i)$. While most of the training data are i.i.d. samples from the same distribution as the target data, a small set of training data is corrupted and assumed to be sampled from a detrimental data distribution $\mathcal{Z}_c$. We denote the detrimental data in the training set by $\mathcal{D}_c$ and the remaining by $\mathcal{D}_l$. Notably, the existence of detrimental data is unknown during data collection and model training.

Due to the existence of detrimental data, the model absorbs unexpected adverse information which adversely affects its generalization ability, resulting in inaccurate predictions. In this paper, we focus on developing a model repair algorithm that aims to improve performance by removing the detrimental information from the data that causes prediction failures. To facilitate model repair, we assume the availability of a small set $\mathcal{D}_r$ of clean data that is reserved for validating the model's success or failure and remains concealed from the model during training. And we identify and remove the detrimental information in the data by leveraging the crucial information from these clean data.

Similar to [33], we formulate the process of model repairment as two steps:

- Step 1: Cause Identification. Given a reserved set $\mathcal{D}_r$ from the target distribution, identify a set of detrimental data points in the training set $\mathcal{D}$ that contribute to the prediction failures for data in $\mathcal{D}_r$.

- Step 2: Model Treatment. Given the identified failure cause set, denoted by $\hat{\mathcal{D}}_c$, refine the model to effectively correct the prediction failures in the target distribution while preserving the performance on remaining

### 2.1 Step 1: Cause Identification

In order to detect the detrimental data points in the training set, it is necessary to first define a measure that quantifies the extent to which a subset of training data contributes to the failed predictions in the reserved set $\mathcal{D}_r$. Then, based on this measure, we can screen out the data points with high failure contributions as the identified data.

Before introducing the measure, it is important to note that the data from detrimental dataset $\mathcal{D}_c$ lead to generalization failure because they introduce a mismatch between train and test distribution. This is aligned with the generalization theory for mixture distributions [17, 38] which asserts that the degree of divergence between the two distributions impacts the generalization error bound. Additionally, the theory demonstrates that an increase in the amount of data from the target distribution during training reduces the generalization bound and decreases the loss of data from the same distribution. Besides, increasing the amount of target distribution data also enables the learned model to fit more closely to the target distribution and less to the detrimental distribution, thereby increasing the loss of detrimental data in $\mathcal{D}_c$. Since the reserved set $\mathcal{D}_r$ is drawn from the target distribution, we can train the target-distribution-enhanced model, denoted $\hat{\phi}$ on the combined set of $\mathcal{D}$ and $\mathcal{D}_r$, and identify the data from $\mathcal{D}_c$ based on the change of loss between the original trained model $\hat{\theta}$ and the enhanced model $\hat{\phi}$. The measure is defined as

$$s(z; \hat{\theta}, \hat{\phi}) = \mathcal{L}(z; \hat{\phi}) - \mathcal{L}(z; \hat{\theta}). \tag{1}$$

By estimating $s(z; \hat{\theta}, \hat{\phi})$ for each data point $z \in \mathcal{D}$, we can identify the data points responsible for the failure, which have a positive value of $s(z; \hat{\theta}, \hat{\phi})$. If we know the size of the cause set, we can identify the cause data by selecting the top $|\mathcal{D}_c|$ data points with the largest $s(z; \hat{\theta}, \hat{\phi})$ values. We denote the identified data set by $\hat{\mathcal{D}}_c$, and the remaining data set by $\hat{\mathcal{D}}_l$.

### 2.1.1 Obtaining the Enhanced Model $\hat{\phi}$

Considering the large scale of data, training a new model $\hat{\phi}$ from scratch is computationally prohibitive. Instead, a practical solution is to approximate the model trained on the combined dataset of $\mathcal{D}$ and $\mathcal{D}_r$ by performing fine-tuning or continual learning on the well-trained $\hat{\theta}$ from $\mathcal{D}$ using the data in $\mathcal{D}_r$. However, due to over-parameterization of the network and the small size of the reserved set $\mathcal{D}_r$, a fine-tuned model may simply memorize the data in $\mathcal{D}_r$ [41], resulting in a negligible impact on the

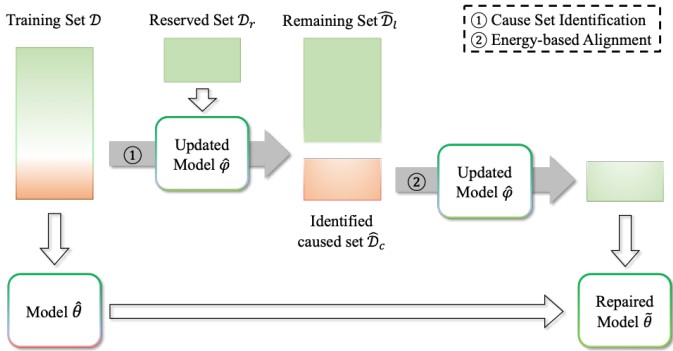

Figure 1: The framework of the proposed method. Best view in color: beneficial and detrimental information are indicated in green and red.

loss of the data in the original training set $\mathcal{D}$. This makes the increase of loss $s(z; \hat{\theta}, \hat{\phi})$ ineffective in identifying the detrimental data. On the other hand, a fine-tuned model may completely overfit $\mathcal{D}_r$ while forgetting the knowledge acquired from $\mathcal{D}$ [27], which can cause it to fail in approximating $\hat{\phi}$.

Given that different layers in a neural network capture distinct patterns [40], selectively updating a subset of layers while freezing the remaining layers can update the detrimental knowledge from $\mathcal{D}_c$ to effective knowledge in $\mathcal{D}_r$, while preserving the acquired knowledge from $\mathcal{D}$. After updating the conflict knowledge from $\mathcal{D}_c$, the new model may result in a more significant increase in loss on $\mathcal{D}_c$. Recent studies [27, 15] support the layer-selective update by observing that selectively fine-tuning a subset of layers improves the fine-tuning generalization for different distribution shifts towards the target distribution. And we empirically validate the effectiveness of layer-selective update in acquiring the knowledge from $\mathcal{D}_r$ and improving the identification accuracy, as presented in Appendix A.4.

To enable layer-selective update, a criterion for layer selection is required. Note that neural networks have a tendency to minimize loss by learning data patterns in the early epochs, but may end up memorizing the data in the later epochs [4, 9]. Therefore, we assume that the reduction in loss after the first epoch can serve as an indicator of the ability of the selected layers to learn data patterns. Based on this assumption, we estimate the loss after performing a one-epoch update for each subset of layers and select the subset with the smallest loss for further updates. Subsequently, the selected subset is updated while the remaining layers are kept frozen. As the reserved set is small, the criterion for layer selection via loss estimation after a one-epoch update is computationally efficient. It should be noted that in practice, we consider only a pre-defined set of layer subsets, rather than all possible combinations of subsets.

**Connection to existing works:** EWC-influence, as proposed in [33], utilized the Bayesian view of continual learning to identify the cause data. Specifically, it defined the identification metric as the drop in log-likelihood between a model trained on the combined dataset $\mathcal{D}$ and $\mathcal{D}_f$, which is a subset of $\mathcal{D}_r$ containing examples that are mispredicted by $\hat{\theta}$, and the current model $\hat{\theta}$ trained on $\mathcal{D}$. By using negative log-likelihood as the loss function, replacing $\mathcal{D}_r$ with $\mathcal{D}_f$, and applying the continual learning algorithm Elastic Weight Consolidation (EWC) [19] to update all layers, our proposed identification measure recovers the EWC-influence introduced in [33].

Both EWC-influence and the proposed method measure the contribution of a training point to the prediction failure by examining the change in the loss. However, the two measures are derived from different purposes. EWC-influence aims to correct the prediction in $\mathcal{D}_f$ of using the gain log-likelihood improvement estimated by deleting the data from $\mathcal{D}$, whereas the proposed method aims to improve the generalization to the data from the same distribution of $\mathcal{D}_r$ by eliminating detrimental information from $\mathcal{D}$. Consequently, the proposed method proposes to measure the contribution to failure by assessing the conflict between $\mathcal{D}$ and $\mathcal{D}_r$ and update the model using all data in $\mathcal{D}_r$.

Linear influence [20, 36, 23] is another effective metric to identify the training points that are most responsible for certain predictions. As analyzed in [33], the linear influence is the same as EWC-influence with the exception that it updates the model via a single update of natural gradient descent. Consequently, our proposed method can recover the linear influence in a similar manner.

## 2.2 Step 2: Model Treatment

In step 1, we identify a partition of the training data $\hat{\mathcal{D}}_c$ that may contain harmful information leading to a suboptimal performance to the test distribution $\mathcal{Z}$. In this section, we present the second step of our method, which involves refining the model using the identified cause set $\hat{\mathcal{D}}_c$ and the trained model $\hat{\theta}$ in conjunction with the enhanced model $\hat{\phi}$. A straightforward solution for repairing the model is to eliminate the harmful information in $\hat{\mathcal{D}}_c$ by removing the identified data from the training set by retraining the model using the remaining data [12] or performing machine unlearning [29] on the identified data [33]. However, identifying a perfect detrimental dataset from the training data is difficult. Simply erasing the information of identified data can negatively impact the model's performance, especially when accurate data is mistakenly identified as detrimental and removed. Furthermore, even the detrimental data contain useful information. For example, an incorrectly annotated image contains a clean input. Recall that the generalization failure is caused by the mismatching between the detrimental distribution and target distribution. Inspired by the domain adaptation framework [35, 30], we propose to align the detrimental data with the target distribution, which can reduce the divergence between the training distribution and the target distribution. The aligning process can rectify the detrimental information in the data, making it possible to reuse the aligned data toward better performance. A theoretical analysis of the following Gaussian mean estimation task supports the potential of this approach to improve performance.

**Gaussian mean estimation task:** Consider the task where the objective is to estimate the mean of a Gaussian distribution $\mathcal{U} = \mathcal{N}(\mu, \sigma^2)$ based on a training set $\mathcal{D}$ consisting of $n$ training data $\{x_i\}_{i=1}^n$ sampled from the distribution $\mathcal{U}$ as well as $m$ training data $\{x_i\}_{i=n+1}^{n+m}$ sampled from another Gaussian distribution $\mathcal{U}' = \mathcal{N}(\mu', \sigma^2)$, where $\mu$ and $\mu'$ are the means of the two Gaussian distributions and $\sigma^2$ denotes the variance. We define the loss function as $\mathcal{L}(x; \theta) = (\theta - x)^2$. In this example, we assume $n$ is sufficiently large than $m$ and the distribution divergence between $\mathcal{U}$ and $\mathcal{U}'$, meausured by $|\mu - \mu'|$, is sufficiently large, i.e. $|\mu - \mu'| > \frac{m+n}{m}\alpha(\sqrt{1/n} - \sqrt{1/(n+m)})$, such that the training data from $\mathcal{U}'$ hurt the generalization performance of the model.

The optimal solution is $\theta^* = \mu$. The empirical risk minimization solution is obtained by minimizing $\mathcal{L}(\mathcal{D}; \theta) = \frac{1}{n+m}\sum_{i=1}^{n+m}(\theta - x_i)^2$ as $\hat{\theta} = \frac{1}{m+n}\sum_{i=1}^{n+m} x_i$. We measure the generalization error as the difference between the empirical solution $\hat{\theta}$ and the optimal solution $\theta^*$. Using Gaussian tail bounds [34], with probability at least $1 - p$ for some probability, the generalization error is upper bounded as

$$|\hat{\theta} - \theta^*| \leq \alpha\sqrt{1/(n+m)} + \frac{m}{m+n}|\mu - \mu'| := B, \tag{2}$$

where $\alpha = \sigma\sqrt{2\log(2/p)}$ is some constant.

Assume we have identified the failure cause set as $\{x_i\}_{i=n+1}^{n+m}$. Directly removing the cause set and retraining the model achieves a generalization error bound as $B_{\text{remove}} = \alpha\sqrt{1/n} < B$.

**Proposition 2.1.** *In the Gaussian mean estimation task, there exists a set of alignment functions:* $h(x) = x - \mu' + \tilde{\mu}$ *where* $|\mu - \tilde{\mu}| << |\mu - \mu'|$ *such that the optimal empirical solution on the combined set of* $\{x_i\}_{i=1}^n$ *and* $\{h(x_i)\}_{i=n+1}^{n+m}$, *i.e.,* $\tilde{\theta} = \frac{1}{m+n}\left(\sum_{i=1}^n x_i + \sum_{i=n+1}^{n+m} h(x_i)\right)$, *achieves a generalization bound* $B_{align} < B_{remove}$.

The detailed proofs for the bound can be found in Appendix B. The alignment function $h(x) = x - \mu' + \tilde{\mu}$ aligns the data in $\mathcal{U}'$ with $\mathcal{U}$, removing the detrimental information $\mu'$ and enhancing the beneficial information $\tilde{\mu}$. Upon alignment, the resulting data set, $\{h(x_i)\}_{i=n+1}^{n+m}$, approximates the target distribution closely, and utilizing it in training will enhance generalization performance.

We extend the idea of alignment-based model treatment from the Gaussian mean estimation task to general tasks. Concretely, we propose to perform alignment on the identified cause data to obtain the aligned data using an Energy-Based Model (EBM) [25], building upon the energy-based alignment proposed in [18]. EBMs represent a probability density $p(z)$ for $z \in \mathbb{R}^d$ as $p_\theta(z) = \frac{\exp(-E_\theta(z))}{Z(\theta)}$, where $E_\theta(z)$ is the energy function which takes $z$ as input and returns a scalar as energy, and $Z(\theta) = \int_z \exp(-E_\theta(z))$ is the normalizing constant. To align a given data point $z'$ with $p_\theta(z)$, we iteratively adapt $z'$ to minimize the energy function [18] as

$$z' = z' - \eta\nabla_{z'}E_\theta(z'),$$

---

**Algorithm 1** Beneficial Information Retaining (BIR) Model Repair

---

1: **Input:** training data $\mathcal{D}$, reserved data $\mathcal{D}_r$; trained model $\hat{\theta}$, a pre-defined candidate set of layer subsets.
   *# Step 1: Cause Identification*
2: Estimate the loss after a one-epoch update of $\hat{\theta}$ on $\mathcal{D}_r$ for each layer subset in the pre-defined set while fixing the remaining layers
3: Select the corresponding layer subset to be updated with the smallest loss
4: Obtain $\hat{\phi}$ by updating the selected layers of model $\hat{\theta}$ on $\mathcal{D}_r$
5: Estimate $s(z; \hat{\theta}, \hat{\phi})$ for each data $z \in \mathcal{D}$
6: Identify the failure cause set as $\hat{\mathcal{D}}_c \leftarrow \{z \in \mathcal{D} : s(z; \hat{\theta}, \hat{\phi}) > 0\}$
   *# Step 2: Model Treatment*
7: Perform data alignment for data in $\hat{\mathcal{D}}_c$ and obtain the aligned cause set $\tilde{\mathcal{D}}_c$
8: Repair the model by updating the selected layers of model $\hat{\theta}$ on $\tilde{\mathcal{D}}_c$

---

where $\eta$ is a hyperparameter that controls the adaptation rate. In contrast to existing works [18] which trains an extra EBM to perform alignment, we adopt the approach proposed in [7], which utilizes the classification model $\theta$ as an energy-based model (EBM) to represent the data distribution density as $p_\theta(x, y) = \frac{\exp(f_\theta(x)[y])}{Z(\theta)}$, where $f_\theta$ is a classification model which maps each data point $x$ to the logits of $K$ classes, $f_\theta(x)[y]$ denotes the logit corresponding to the $y$-th class and $Z(\theta)$ is the unknown normalization constant.

We use target-distribution-enhanced model $\hat{\phi}$ to approximately characterize the distribution $\mathcal{Z}$. Considering potential label noise, we update $x$ using the energy function of $p_{\hat{\phi}}(x)$ instead of $p_{\hat{\phi}}(x, y)$, where we obtain $p_{\hat{\phi}}(x) = \sum_{y'} p_{\hat{\phi}}(x, y') = \frac{\sum_{y'} \exp(f_{\hat{\phi}}(x)[y'])}{Z(\hat{\phi})}$ by marginalizing out $y$. The corresponding energy function is defined as

$$E_{\hat{\phi}}(x) = -\log \sum_{y'} \exp(f_{\hat{\phi}}(x)[y']). \tag{3}$$

With the energy function, we align the data $(x, y) \in \hat{\mathcal{D}}_c$ by first aligning the input $x$ by one-step gradient descent as

$$\tilde{x} = x - \eta \nabla_x E_{\hat{\phi}}(x), \tag{4}$$

then aligning the label $y$ by

$$\tilde{y} = p_{\hat{\phi}}(y|\tilde{x}) = \frac{\exp(f_{\hat{\phi}}(\tilde{x}))}{\sum_{y'} \exp(f_{\hat{\phi}}(\tilde{x})[y'])}, \tag{5}$$

which is the soft label of $\tilde{x}$.

After obtaining the aligned data, we update the model from $\hat{\theta}$ using the aligned data. Similar to the update of enhanced model $\hat{\phi}$ we update only the layers selected in the cause identification step while keeping the remaining layers frozen.

Finally, the proposed model repair method, named Beneficial Information Retaining (BIR) Model Repair, including the cause identification and model refinement, is summarized in Algo. 1 and illustrated in Fig. 1.

## 3 Related Works

Our work is closely related to [33], following which we formulate the model repairment problem into a cause identification step and a model treatment step. In [33], Tanno et al. discuss the model repairment problem from a Bayesian perspective and introduce EWC-influence for cause identification and EWC-deletion for model treatment which performs continual unlearning to remove the data. The cause identification relies on data influence estimation, which can be solved by various influence estimation methods [10] such as linear influence [20, 22], SGD influence [12] and Data Shapley Value [6, 16]. For model treatment, existing works [12, 23] proposes to remove the information of the failure cause data by retraining the model with the remaining data or by performing machine unlearning [8, 29]. RDIA [23] proposes an influence-based relabeling framework for reusing harmful training samples. However, RDIA's aggressive relabeling of benign data that is mistakenly identified as detrimental can introduce incorrect information and negatively impact the model's performance,

Table 1: Comparison of model repair performance on corrupted MINST datasets. Performance better than base model is highlighted by cyan and the best performance among all baselines is in bold.

| Model | Dataset | MNIST-Label | | | MNIST-Input | | | MNIST-Adv | | |
|---|---|---|---|---|---|---|---|---|---|---|
| | Metric | All | Clean | Noisy | All | Clean | Noisy | All | Clean | Noisy |
| LeNet | Base Model | 82.46 | 96.10 | 62.00 | 96.46 | 97.10 | 95.50 | 96.38 | 97.33 | 94.95 |
| | SGD Influence Repair | 84.40 | 95.67 | 67.50 | 96.12 | 96.90 | 94.95 | 96.06 | 96.97 | 94.70 |
| | Linear Influence Repair | 83.00 | **96.43** | 62.85 | **96.26** | **97.07** | 95.05 | **96.50** | **97.27** | 95.35 |
| | EWC Repair | 84.26 | 95.43 | 67.50 | 96.12 | 96.90 | 94.95 | 96.06 | 96.97 | 94.70 |
| | RDIA | 72.70 | 93.03 | 42.20 | 85.82 | 90.83 | 78.30 | 81.30 | 86.00 | 74.25 |
| | **BIR (Ours)** | **95.60** | 95.93 | **95.10** | 96.16 | 96.20 | **96.10** | 96.44 | 96.47 | **96.40** |
| Conv4 | Base Model | 83.96 | 97.87 | 63.10 | 98.00 | 98.57 | 97.15 | 97.82 | 98.57 | 96.70 |
| | SGD Influence Repair | 88.66 | **98.60** | 73.75 | 97.76 | 98.40 | **96.80** | 97.00 | 97.93 | 95.60 |
| | Linear Influence Repair | 72.12 | 87.97 | 48.35 | 96.96 | 97.43 | 96.25 | 92.86 | **98.43** | 84.50 |
| | EWC Repair | 89.02 | 98.53 | 74.75 | 97.82 | 98.50 | **96.80** | 97.10 | 97.90 | 95.90 |
| | RDIA | 74.04 | 90.03 | 50.05 | 82.76 | 79.77 | 87.25 | 77.96 | 87.17 | 64.15 |
| | **BIR (Ours)** | **96.36** | 97.10 | **95.25** | **98.02** | **98.93** | 96.65 | **98.12** | **98.43** | **97.65** |

Table 2: Comparison of model repair performance on corrupted CIFAR-10 datasets.

| Model | Dataset | CIFAR10-Label | | | CIFAR10-Input | | | CIFAR10-Adv | | |
|---|---|---|---|---|---|---|---|---|---|---|
| | Metric | All | Clean | Noisy | All | Clean | Noisy | All | Clean | Noisy |
| ResNet20 | Base Model | 74.80 | 90.83 | 48.75 | 87.20 | 91.77 | 80.35 | 87.88 | 93.40 | 79.60 |
| | SGD Influence Repair | 76.66 | **92.40** | 53.05 | **89.24** | 93.00 | **83.60** | 86.06 | 92.13 | 76.95 |
| | Linear Influence Repair | 74.80 | 92.30 | 48.55 | 87.88 | 92.83 | 80.45 | 87.72 | **93.00** | 79.80 |
| | EWC Repair | 77.06 | 90.77 | 56.50 | 87.44 | **93.37** | 78.55 | 86.70 | 92.00 | 78.75 |
| | RDIA | 67.50 | 85.17 | 41.00 | 79.66 | 85.20 | 71.35 | 77.54 | 82.23 | 70.50 |
| | **BIR (Ours)** | **83.98** | 91.10 | **73.30** | 87.96 | 91.40 | 82.80 | **87.94** | 92.17 | **81.60** |
| ResNet50 | Base Model | 77.10 | 90.60 | 50.95 | 86.62 | 94.30 | 75.10 | 87.60 | 93.13 | 79.30 |
| | SGD Influence Repair | 78.36 | **94.17** | 54.65 | 89.36 | **95.30** | 80.45 | 84.08 | **94.67** | 68.20 |
| | Linear Influence Repair | 75.82 | 93.83 | 48.80 | 88.28 | 93.90 | 79.85 | 87.60 | 93.67 | 78.70 |
| | EWC Repair | 78.00 | 93.57 | 54.65 | **89.94** | 94.47 | 83.15 | 84.08 | **94.67** | 68.20 |
| | RDIA | 66.42 | 90.13 | 30.85 | 76.24 | 87.33 | 59.60 | 71.44 | 84.20 | 52.30 |
| | **BIR (Ours)** | **83.02** | 91.73 | **69.95** | 88.64 | 91.80 | **83.90** | **90.26** | 93.20 | **85.85** |

Table 3: Comparison of model repair performance on CIFAR-10N. The network architecture is ResNet20.

| | CIFAR-10N | | |
|---|---|---|---|
| | Worst | Random | Aggregate |
| Base Model | 77.76 | 84.68 | 85.78 |
| Linear Influence Repair | 79.30 | 85.22 | 87.70 |
| SGD Influence Repair | 71.50 | 77.21 | 76.08 |
| EWC Repair | 71.80 | 76.22 | 77.18 |
| RDIA | 62.90 | 62.96 | 64.62 |
| **BIR (Ours)** | **85.50** | **88.44** | **89.60** |

making it a high-risk approach in the case of imperfect detrimental data identification. Our proposed approach BIR also reuses the identified data to improve model performance. By aligning the identified detrimental dataset with the clean data, our framework rectifies the detrimental information while preserving the beneficial information in the identified set, and updates the model using the aligned data. Unlike RDIA, our alignment strategy does not introduce incorrect information when applied to clean data and achieves satisfactory performance even in cases where there are mistakenly identified clean data in the cause set.

## 4 Experiments

To evaluate the effectiveness of the proposed BIR model repair algorithm, we conduct extensive experiments to answer the following research questions: **Q1**: How does BIR compare to the state-of-the-art baselines in repairing neural networks that produce error predictions? **Q2**: How does the proposed failure cause data identification metric compare to the baselines in identifying the failure cause data? **Q3**: How does the proposed align-then-update strategy compare to the baselines in model treatment when considering both perfect and imperfect failure cause sets?

**Datasets and Network Architectures:** We evaluate the effectiveness of model repair algorithms on corrupted versions of MNIST [26] and CIFAR-10 [24] datasets following the setups in [33]. We consider annotation error, input noise, and adversarial attacks in the training data. To simulate annotation error, we randomly flip about 40% of labels between specific similar digits or classes. For input noise, we randomly corrupt 40% of MNIST images and 30% of CIFAR-10 images in a selected set of target classes by adding specific types of noise, such as salt-and-pepper in MNIST or Gaussian noise in CIFAR-10. Additionally, we construct datasets by adversarially attacking 40% of randomly selected training images in the same target classes using FGSM. We denote the corrupted datasets with label noise, input noise, and adversarial attacks by MNIST-Label, MNIST-Input, and MNIST-Adv

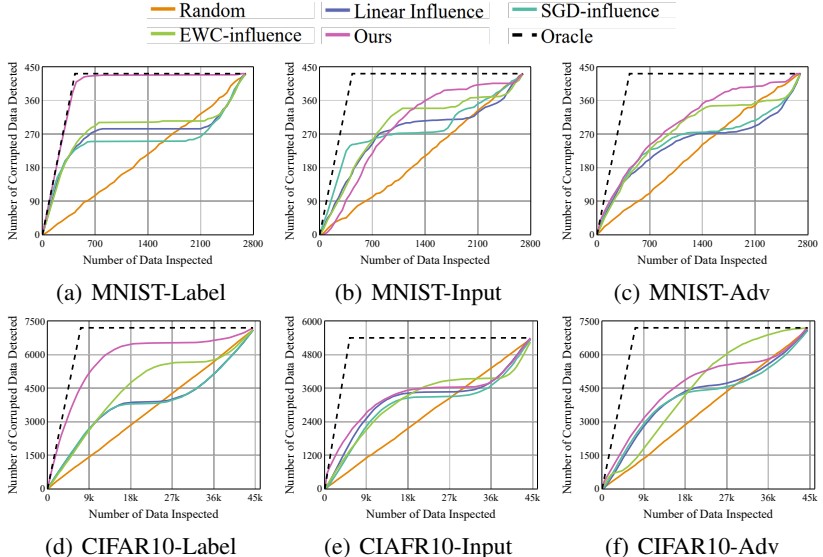

| (a) MNIST-Label | (b) MNIST-Input | (c) MNIST-Adv |
| (d) CIFAR10-Label | (e) CIAFR10-Input | (f) CIFAR10-Adv |

Figure 2: Comparison of Failure Cause Identification on LeNet trained with corrupted MNIST datasets and ResNet50 trained with CIFAR-10.

(CIFAR10-Label, CIFAR10-Input, and CIFAR10-Adv). For MNIST, 3000 samples from the original training set are utilized while the complete original training dataset of CIFAR-10 is employed to train the base models, with a validation set consisting of 10% of the data. Furthermore, we partitioned the original test set, which is uncorrupted, into a reserved set consisting of 300 samples for MNIST and 1000 samples for CIFAR-10, while the remaining set was designated as the testing set. The selected target classes of MNIST are 1,7, 6, and 9; and the selected target classes of CIFAR-10 are 'plane', 'bird', 'cat', and 'dog'. LeNet and Conv4, which is a 4-layer CNN with a fully connected layer, are employed as classifiers for MNIST, while ResNet-20 and ResNet-50 are utilized for CIFAR-10. We also conduct experiments on CIFAR-10N dataset [37], which equips the training set of CIFAR-10 with real-world human-annotated noisy labels. There are 3 noise levels: *Worst* (∼41%), *Random* (∼18%), and *Aggregate* (∼9%).

**Baselines:** We compare the proposed method with EWC Repair [33], RDIA [23], SGD-influence Repair [12] and Linear-influence-based Repair. EWC Repair utilizes EWC-influence to identify harmful data and EWC-deletion to remove them from the model. RDIA uses linear influence [20] to identify causes, relabel the identified data using an influence-based relabeling function, and retrains the model using the relabeled and remaining data. SGD-influence Repair identifies harmful data using SGD-influence and proposes to retrain the model using the remaining data. Linear-Influence Repair uses the linear influence function [20] to identify the cause data and performs Newton-update removal [8] to eliminate the identified data's information from the model. In practice, retraining the model from scratch can be computationally prohibitive. Therefore, for RDIA, we performed fine-tuning by using both the relabeled data and an equivalent amount of randomly selected data from the remaining dataset. For SGD-influence Repair, fine-tuning was conducted on a small partition of the remaining data that matched the size of the identified data.

**Performance Measure**: We measure the performance of model repair by the accuracy of the entire test set (denoted by "Test"). Besides, we measure the success rate of the repair by the test accuracy of the selected corrupted classes (denoted by "Noisy") and the performance maintaining rate by the test accuracy of the remaining classes (denoted by "Clean").

Additional information regarding the setups

## 4.1 Comparison to Model Repairment Baselines

The comparison results for corrupted MNIST and CIFAR-10 are provided in Tabs. 1 and 2. We can observe from the results that the proposed BIR repair algorithm successfully repairs the model and improves the performance of noisy classes in all experiments except for MNIST-Input with Conv4. In contrast, most baselines fail to repair the model for MNIST-Input, MNIST-Adv and CIFAR10-Adv.

Table 4: Comparison of Model Treatment for LeNet trained on corrupted MNIST.

| Dataset | MNIST-Label | | | MNIST-Input | | | MNIST-Adv | | |
|---|---|---|---|---|---|---|---|---|---|
| Metric | Test All | Test Clean | Test Noisy | Test All | Test Clean | Test Noisy | Test All | Test Clean | Test Noisy |
| Base Model | 82.46 | 96.10 | 62.00 | 96.46 | 97.10 | 95.50 | 96.38 | 97.33 | 94.95 |
| Retrain | 96.66 | 97.43 | 95.50 | 96.84 | 97.57 | 95.75 | 96.96 | 97.53 | 96.10 |
| Identified set = ground-truth cause set | | | | | | | | | |
| Fine-tune | 95.56 | 96.90 | 93.55 | 96.42 | **97.17** | 95.30 | 96.26 | 97.17 | 94.90 |
| Newton Removal | 85.20 | 96.33 | 68.50 | 96.52 | **97.17** | 95.55 | 96.50 | **97.33** | 95.25 |
| EWC-Deletion | 95.10 | 96.00 | 93.75 | 96.14 | 96.50 | 95.60 | 96.24 | 96.57 | 95.75 |
| RDIA | **95.70** | **97.23** | 93.40 | 95.94 | 96.63 | 94.90 | 92.50 | 92.67 | 92.25 |
| BIR (Ours) | 95.64 | 95.67 | **95.60** | **96.78** | 97.11 | **96.30** | **96.66** | 97.13 | **95.95** |
| Identified set = 75% ground-truth cause set ∪ 25% clean data | | | | | | | | | |
| Fine-tune | 88.18 | 96.23 | 76.10 | 95.98 | 96.37 | 95.40 | 96.22 | 96.53 | 95.45 |
| Newton Removal | 83.92 | **96.47** | 65.10 | 96.16 | **96.87** | 95.10 | 96.36 | **97.30** | 94.95 |
| EWC-Deletion | 87.74 | 94.70 | 77.30 | 95.74 | 96.40 | 94.75 | 95.46 | 96.03 | 94.60 |
| RDIA | 84.86 | 83.77 | 86.50 | 90.48 | 90.07 | 91.10 | 85.82 | 89.40 | 80.45 |
| BIR (Ours) | **95.70** | **96.47** | 94.55 | 96.48 | 96.67 | **96.19** | 96.62 | 96.80 | **96.35** |

Table 5: Comparison of Model Treatment for ResNet50 trained on corrupted CIFAR-10.

| Dataset | CIFAR10-Label | | | CIFAR10-Input | | | CIFAR10-Adv | | |
|---|---|---|---|---|---|---|---|---|---|
| Metric | Test All | Test Clean | Test Noisy | Test All | Test Clean | Test Noisy | Test All | Test Clean | Test Noisy |
| Base Model | 77.10 | 90.60 | 50.95 | 86.62 | 94.30 | 75.10 | 87.60 | 93.13 | 79.30 |
| Retrain | 88.18 | 94.03 | 79.40 | 87.48 | 91.43 | 81.55 | 88.10 | 93.07 | 80.65 |
| Identified set = ground-truth cause set | | | | | | | | | |
| Fine-tune | **85.16** | **94.83** | 70.65 | **89.86** | **94.40** | 83.05 | 89.66 | **94.67** | 82.15 |
| Newton Removal | 84.04 | 93.60 | 69.70 | 86.72 | 92.60 | 77.90 | 87.60 | 93.80 | 78.30 |
| EWC-Deletion | 84.50 | 94.73 | 69.15 | 89.76 | 94.33 | 82.90 | 88.42 | 94.20 | 79.70 |
| RDIA | 82.64 | 94.43 | 64.95 | 87.20 | 92.07 | 79.90 | 86.98 | 92.40 | 78.85 |
| BIR (Ours) | 83.26 | 91.43 | **71.00** | 88.84 | 91.97 | **84.15** | **90.60** | 93.43 | **86.10** |
| Identified set = 75% ground-truth cause set ∪ 25% clean data | | | | | | | | | |
| Fine-tune | **83.18** | **94.47** | 66.25 | **89.00** | **93.50** | 82.25 | 89.48 | **94.57** | 81.85 |
| Newton Removal | 76.08 | 94.17 | 48.95 | 86.72 | 91.50 | 79.45 | 87.88 | 93.00 | 80.20 |
| EWC-Deletion | 82.08 | 94.07 | 64.10 | 88.84 | 93.27 | 82.20 | 87.12 | 92.83 | 78.55 |
| RDIA | 74.58 | 86.17 | 57.20 | 79.88 | 85.30 | 71.65 | 78.70 | 83.57 | 71.40 |
| BIR (Ours) | 82.88 | 91.33 | **70.20** | 88.78 | 92.83 | **83.35** | **90.50** | 93.63 | **86.05** |

Furthermore, BIR achieves the best performance in noisy classes on 10/12 experiments, and the best performance in all classes on 8/12 experiments. These results demonstrate the effectiveness and superiority of BIR in removing detrimental information from the data to repair the model. We also observe that performance in clean classes of all baselines is generally slightly lower than the base model, which is a trade-off for repairing the corrupted data. The comparison results for CIFAR-10N in Tab. 3 show that our method achieves superior performance across all noise levels, demonstrating its effectiveness in mitigating real-world label noise.

## 4.2 Comparison of Failure Cause Identification

We compare the proposed causes identification method using Eqn. (1) with EWC-influence [33], Linear-influence [20] and SGD-influence [12]. To measure the performance of the failure cause identification, we report the number of identified data points corresponding to samples corrupted with noise. The results in Figure 2 show that all methods perform well when a small number of data points are identified. Our method performs the best overall, except on MNIST-Input. It is worth noting that our method identifies almost all the corrupted data points in MNIST-Label and CIFAR10-Label, where the model performance is significantly impacted by the detrimental information in the training data. However, it is also important to note that the identification is not perfect, and a sufficiently large set of clean data points may be identified as detrimental data. Therefore, the model repair baselines are at risk of removing useful information from the identified set and may fail to repair the model. In contrast, BIR can effectively remove the detrimental information while retaining beneficial information from the identification, achieving satisfactory performance.

## 4.3 Comparison of Model Treatment

We evaluate model treatment algorithms given both precise and imprecise identified cause sets, with the latter containing 75% corrupted data and 25% clean data. The baselines include EWC-deletion [33], RDIA [23], and Newton-update removal [8] and fine-tuning of randomly selected data from remaining set as baselines. In addition, we employ retraining the model from scratch using the remaining data as a further baseline.

The results are reported in Tabs. 4 and 5. We can observe performance improvement compared to the results reported in Tabs. 1 and 2. And we also observe a decline in performance for most methods

when replacing the precise identified set with the imprecise one, particularly on MNIST-Label and CIFAR10-Label. These results indicate that the performance of the model treatment baselines is contingent on the correctness of identified set. Notably, the performance drop of BIR is significantly less than that of the baselines, illustrating the effectiveness and superiority of BIR in retaining beneficial information and eliminating harmful information from the identified data to repair the model, even when the identified set comprises clean data.

## 4.4   Extra Experiment Results

We investigate the effect of the layer-selective update strategy on cause identification and model treatment. The results and analysis are presented in Appendix A.4. Moreover, we explore the solution for determining the appropriate size of the cause set when the ground-truth size is unknown and evaluate the effect of the size on performance in Appendix A.5. Additionally, in Appendix A.6, we studied the effect of varying the amount of clean reserved data on our method's performance.

## 5   Conclusion and Limitations

In this paper, we propose a novel framework for repairing neural networks trained with detrimental data. To identify the data that cause generalization failure, we propose to update the model with a reserved set sampled from the target distribution, which reduces the loss of clean data and increases the loss of corrupted data. After identifying the cause data, we acknowledge the potential value of detrimental data in enhancing the training of a more generalized model alongside target data and propose an energy-based alignment to remove harmful information while retaining beneficial information in the identified data. After obtaining the aligned data, we update the model on these data to repair the model. We conducted extensive experiments on real-world datasets to validate the effectiveness of our proposed framework, demonstrating its superiority over existing methods in identifying harmful data and repairing the model to enhance generalization.

Limitation: the proposed method entails access to the training data for model repairing purposes, a requirement that may be hampered by the possibility of discarding the data after training. Nonetheless, such accessibility is essential to the method's development, which is predicated on the assumption that erroneous training data engenders generalization failure, and thus seeks to remove the adverse information to effect model repair. We leave the investigation of training data-free model repair to the future.

## Acknowledgements

Sinno J. Pan thanks for the support from HK Global STEM Professorship and the JC STEM Lab of Machine Learning and Symbolic Reasoning.

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

# A Omitted Experiment Details in Sec. 4 and Extra Experiments

## A.1 Dataset

**MNIST** [26] dataset contains 60,000 training images and 10,000 test images of 10-digit classes from 0 to 9, with each image in the size $28 \times 28$. We follow [33] to randomly select 3,000 images from the original training set to construct the training set. And we randomly choose 300 images from the original test set to construct the reserved set and use the remaining data as the test set for performance evaluation.

We construct three corrupted versions of MNIST dataset by introducing label noise, input noise, and adversarial attack on the clean training set. Specifically, for **MNIST-Label**, we randomly select 40% of the data from classes 1 and 7, and flipped their labels to each other; and the same for classes 6 and 9. For **MNIST-Input**, we randomly corrupted 40% of the data from the target classes 1, 7, 6, and 9 using salt and pepper noise. To create **MNIST-Adv**, we first train a victim model on the clean training set and then generate adversarial images using the fast gradient sign method (FGSM) for 40% of the data randomly selected from the target classes. The poisoned images in MNIST-Adv are labeled by the classes predicted by the victim model.

**CIFAR-10** [24] comprises 60,000 color images divided into 10 categories, each with dimensions of $32 \times 32$. The training set has 50,000 images while the original test set is split into a reserved set with 1,000 images and a test set with 9,000 images. We introduce three corrupted versions of the CIFAR-10 dataset, where label noise, input noise, and adversarial attacks are introduced on the target classes: plane, bird, cat, and dog. Specifically, we generate the **CIFAR10-Label** dataset by randomly selecting 40% of the plane and bird class images and swapping their labels, and similarly for the cat and dog classes. For **CIFAR-Input**, we apply Gaussian noise to 30% of the images from the target classes. Finally, for **CIFAR10-Adv**, we use adversarial attacks to corrupt 40% of the randomly selected images from the target classes, similar to MNIST-Adv.

**CIFAR-10N** [37] is CIFAR-10 dataset with human-annotated noisy labels. In CIFAR-10N dataset, each training image has 3 human-annotated labels. It has 5 noisy-label sets: **Random** $i(i \in \{1, 2, 3\})$ uses the $i$-th submitted label for each image; **Aggregate** generates the label by majority voting of three noisy labels; **Worst** randomly selected one label from the wrong labels if there exist any wrongly annotated labels in 3 noisy labels. In the experiments, we adopt 3 out of the 5 noisy-label sets, namely, *Aggregate*, *Random 2* and *Worst*.

## A.2 Network Architectures

For the corrupted MNIST datasets, we use LeNet and Conv4 as the classifier. LeNet is widely used for the tasks on MNIST. It consists of 2 convolutional layers and 3 fully-connected layers. In the layer selective update, we define Layer 1 to Layer 5 as the first (the first convolutional layer) to the last layer (the last fully-connected layer). The set of candidate layer subsets considered in the layer selective update stage comprises {{Layer 1}, {Layer 2}, {Layer 3}, {Layer 4}, {Layer 5}, {Layer 1-2}, {Layer 1-3}, {Layer 1-5}}. Conv4 consists of 4 convolution layers followed by a fully-connected layer. Similar to LeNet, we define Layer 1 to Layer 5 as the first to the last layer, respectively, and consider the same candidate layer subsets.

The corrupted CIFAR-10 datasets are classified using ResNet-20 and ResNet-50. The CIFAR-10N dataset also uses ResNet-20 as classifier. ResNet-20 comprises one convolutional layer, 3 compositional layers, each of which consists of 3 basic blocks, and a fully-connected layer. Layer 1 is defined as the first convolutional layer, Layers 2 to 4 are the 3 compositional layers, and Layer 5 is the fully-connected layer. The same candidate layer subsets are considered. ResNet-50 consists of one convolutional layer, 4 compositional layers, each of which consists of 3, 4, 6, and 3 bottleneck blocks, respectively, and a fully-connected layer. Layer 1 is the first convolutional layer, Layers 2 to 5 are the 4 compositional layers, and Layer 6 is the last fully-connected layer. Its candidate layer subsets are defined as {{Layer 1}, {Layer 2}, {Layer 3}, {Layer 4}, {Layer 5}, {Layer 6}, {Layer 1-2}, {Layer 1-3}, {Layer 1-6}}. For ResNet-50, The input images are resized to $224 \times 224$.

LeNet and Conv4 are both trained from scratch. The ResNet-20 and ResNet-50 model is initialized with the pre-trained model on ImageNet.

Table 6: Performance with different selective layers updated: Identification AUC and Treatment Accuracy(%), with the best result highlighted in bold. The corresponding performance of the selected layers update is highlighted in gray. The backbone of the classifier is LeNet.

| Dataset | MNIST-Label | | | MNIST-Input | | | MNIST-Adv | | |
|---|---|---|---|---|---|---|---|---|---|
| | Loss after one-epoch update | Identification AUC | Treatment Accuracy(%) | Loss after one-epoch update | Identification AUC | Treatment Accuracy(%) | Loss after one-epoch update | Identification AUC | Treatment Accuracy(%) |
| Layer 1 | 1.0795 | 0.91 | 86.02 | 0.2503 | **0.71** | **96.16** | 0.2418 | 0.61 | **96.54** |
| Layer 2 | 0.8967 | 0.96 | 92.56 | 0.2641 | 0.61 | 96.14 | 0.2201 | 0.57 | 96.40 |
| Layer 3 | 0.8145 | 0.98 | 95.36 | 0.4268 | 0.60 | 95.92 | 0.3458 | 0.60 | 96.28 |
| Layer 4 | 0.9419 | 0.96 | 94.42 | 0.3027 | 0.52 | 95.78 | 0.2682 | 0.56 | 96.34 |
| Layer 5 | 1.0725 | 0.96 | 93.14 | 0.2751 | 0.46 | 95.52 | 0.2289 | 0.51 | 96.18 |
| Layer 1-2 | 0.9193 | 0.96 | 92.36 | 0.2799 | 0.58 | 96.14 | 0.2121 | 0.70 | 96.44 |
| Layer 1-3 | 0.7360 | **0.99** | **95.60** | 0.4243 | 0.70 | 96.12 | 0.3311 | **0.73** | 96.46 |
| Layer 1-5 | 0.9407 | **0.99** | 95.50 | 0.6009 | 0.69 | 95.34 | 0.3618 | 0.66 | 96.28 |

Table 7: Performance with different selective layers updated: Identification AUC and Treatment Accuracy(%), with the best result highlighted in bold. The corresponding performance of the selected layers update is highlighted in gray. The backbone of the classifier is ResNet20.

| Dataset | CIFAR10-Label | | | CIFAR10-Input | | | CIFAR10-Adv | | |
|---|---|---|---|---|---|---|---|---|---|
| | Loss after one-epoch update | Identification AUC | Treatment Accuracy(%) | Loss after one-epoch update | Identification AUC | Treatment Accuracy(%) | Loss after one-epoch update | Identification AUC | Treatment Accuracy(%) |
| Layer 1 | 1.0763 | 0.56 | 74.82 | 0.6490 | 0.68 | 85.52 | 0.6438 | 0.60 | 87.64 |
| Layer 2 | 1.0520 | 0.57 | 75.00 | 0.5865 | **0.84** | **87.96** | 0.6058 | 0.69 | 86.60 |
| Layer 3 | 1.0038 | 0.68 | 76.92 | 0.6489 | 0.70 | 86.46 | 0.6035 | 0.65 | 86.58 |
| Layer 4 | 0.7813 | **0.87** | **83.98** | 0.6445 | 0.74 | 86.78 | 0.5936 | 0.68 | **87.94** |
| Layer 5 | 0.9988 | 0.60 | 75.62 | 0.6572 | 0.69 | 87.46 | 0.6199 | 0.63 | 87.34 |
| Layer 1-2 | 1.0213 | 0.63 | 74.82 | 0.6479 | 0.80 | 85.92 | 0.6292 | 0.69 | 86.58 |
| Layer 1-3 | 1.0383 | 0.66 | 76.04 | 0.6309 | 0.76 | 86.08 | 0.6080 | 0.65 | 86.72 |
| Layer 1-5 | 0.7947 | 0.86 | 83.70 | 0.6642 | 0.72 | 87.22 | 0.6298 | **0.71** | 86.58 |

## A.3  Training Details

In this section, we introduce the settings in training. Specifically, the training set is partitioned into 90% training data and 10% validation data. The base model is trained using the Adam optimizer with a learning rate of 1e-3 and a batch size of 64 for all datasets. And the model is trained for a maximum of 100 epochs, and early stopping after 5 epochs of no progress on the validation data.

To obtain the enhanced model, we update the model on all data in the reserved set using the Adam optimizer with a learning rate of 1e-3 and a batch size of 16. Training is stopped after no progress on the training loss. The update layers are selected based on the training loss estimated after a one-epoch update, i.e., an update going through all reserved data. Once the enhanced model is obtained, we estimate the score in Eqn. (1) for all training data and identify the cause data by selecting the top $\tilde{K}$ data points with the highest $s(z; \hat{\theta}, \hat{\phi})$ values. Here, $\tilde{K}$ is equal to the size of the ground-truth cause set, i.e., $|\mathcal{D}_c|$. For the baseline identification methods, the same number of data points, i.e. $\tilde{K}$ are selected as the cause data.

During model treatment, we first perform energy-based alignment on the identified data using the enhanced model. Specifically, the data input is updated using Eqn. (4) with $\eta = 1.0$ and the label is updated using Eqn. (5). We then update the model using the aligned data and $\tilde{K}$ additional data points that are randomly sampled from the remaining training set. The purpose of the additional data from the remaining set is to alleviate bias toward the identified data. It is worth noting that all baseline methods perform model treatment using both identified data and randomly sampled remaining data.

In the proposed method, we initialize the model with $\hat{\theta}$ and optimize it using the Adam optimizer with a learning rate of 1e-3. For the corrupted MNIST datasets and CIFAR-10 datasets, we set the batch size to 16 and 64, respectively. We terminate the training process when there is no further decrease in the training loss.

## A.4  Extra Experiment: The Effect of Layer-Selective Update

In this section, we investigate the effect of the layer-selective update strategy on cause identification and model treatment. Performances are presented in Tabs. 6 and 7, where the AUC between the cause score of the training data calculated by Eqn. (1) and the ground-truth label indicating whether the data is from $\mathcal{D}_c$ is used to measure identification performance, and the accuracy of the test set of the repaired model with different layers updated is used to evaluate model treatment performance. The

Table 8: Number of identified cause data on corrupted MNIST and CIFAR-10 datasets using different strategies.

| | MNIST-Label | MNIST-Input | MNIST-Adv | CIFAR10-Label | CIFAR10-Input | CIFAR10-Adv |
|---|---|---|---|---|---|---|
| ground-truth amount | 432 | 432 | 432 | 7200 | 5400 | 7200 |
| threshold = 0 | 1115 | 1789 | 1919 | 23955 | 27449 | 30457 |
| threshold = $\log(1.1)$ | 469 | 122 | 198 | 8745 | 5076 | 6003 |
| threshold = $\log(1.05)$ | 487 | 157 | 251 | 9952 | 6362 | 7686 |
| threshold = $\log(1.01)$ | 531 | 268 | 393 | 12836 | 9716 | 11849 |

Table 9: The repair performance on corrupted MNIST datasets under different cause set sizes using different identified strategies. The size of the identified cause set is available in Tab. 8. The network is LeNet.

| | MNIST-Label | | | MNIST-Input | | | MNIST-Adv | | |
|---|---|---|---|---|---|---|---|---|---|
| | All | Clean | Noisy | All | Clean | Noisy | All | Clean | Noisy |
| ground-truth amount | 95.60 | 95.93 | 95.10 | 96.16 | 96.20 | 96.10 | 96.44 | 96.47 | 96.40 |
| threshold = 0 | 95.84 | 96.30 | 95.15 | 96.32 | 96.13 | 96.60 | 96.48 | 96.63 | 96.25 |
| threshold = $\log(1.1)$ | 95.64 | 95.77 | 95.45 | 96.22 | 96.13 | 96.35 | 96.40 | 96.57 | 96.15 |
| threshold = $\log(1.05)$ | 95.66 | 96.20 | 94.85 | 96.28 | 96.07 | 96.60 | 96.30 | 96.43 | 96.10 |
| threshold = $\log(1.01)$ | 95.56 | 96.07 | 94.80 | 96.14 | 95.97 | 96.40 | 96.34 | 96.53 | 96.05 |

Table 10: The repair performance on corrupted CIFAR-10 datasets under different cause set sizes using different identified strategies. The size of the identified cause set is available in Tab. 8. The network is ResNet50.

| | CIFAR10-Label | | | CIFAR10-Input | | | CIFAR10-Adv | | |
|---|---|---|---|---|---|---|---|---|---|
| | All | Clean | Noisy | All | Clean | Noisy | All | Clean | Noisy |
| ground-truth amount | 83.02 | 91.73 | 69.95 | 88.64 | 91.80 | 83.90 | 90.26 | 93.20 | 85.85 |
| threshold = 0 | 83.26 | 91.33 | 71.15 | 88.90 | 92.20 | 83.55 | 90.14 | 93.43 | 85.20 |
| threshold = $\log(1.1)$ | 83.00 | 91.50 | 70.25 | 88.50 | 91.63 | 83.80 | 90.20 | 93.30 | 85.55 |
| threshold = $\log(1.05)$ | 83.14 | 91.83 | 70.10 | 88.62 | 91.70 | 84.00 | 90.12 | 93.03 | 85.75 |
| threshold = $\log(1.01)$ | 82.98 | 91.57 | 70.10 | 88.66 | 92.07 | 83.55 | 89.86 | 93.07 | 85.05 |

results indicate that updating all layers is suboptimal for both identification and repair performance in most cases. Thus, a layer-selective update is necessary to achieve better performance. We suggest selecting the layers for update based on the loss after one-epoch update on the reserved set. The proposed method achieves the best identification performance in 4/6 cases and the best treatment performance in all datasets except MNIST-Adv, where it is only 0.1% less accurate than the best one. These results highlight the effectiveness of the layer-selective update in enhancing the proposed method's performance.

## A.5 Extra Experiment: The Effect of Identified Cause Set Size

In our experiments, we assume knowledge of the ground-truth size of the cause set, denoted as $N_C$, and identify the cause data by selecting the top $N_C$ data points with the highest score estimated by Eqn. (1). However, in cases where the number of corrupted data examples is uncertain, selecting an appropriate threshold becomes challenging. To address this issue, a practical solution is to apply a fixed threshold on the score ($s$) computed using Eqn. (1) as a heuristic measure. As a straightforward solution, setting the threshold to 0 identifies all training data with an increased loss, but this often results in identifying more data than the actual number of corrupted points, as demonstrated in our empirical studies presented in Tab. 8. Our investigations indicate that a threshold between $\log(1.01)$ and $\log(1.1)$ effectively filters out identified data close in size to the ground truth. Thus, we recommend employing a threshold of $\log(1.05)$ for practical applications.

Importantly, we emphasize that the repair performance of our proposed BIR method is robust to the size of the identified cause set, as evidenced by the experimental results in Tabs. 9 and 10. Therefore, while identification using thresholds may not be perfect, the repair performance of BIR is hardly affected.

## A.6 Extra Experiment: The Effect of Clean Reserved Set Size

In this section, we investigate the impact of varying the amount of reserved clean data on the performance of the proposed model repair method when applied to three corrupted MNIST datasets. The identification and repair performance is reported in Tab. 11. Our empirical analysis reveals three key findings: firstly, with a mere 30 clean samples, our approach resulted in enhanced performance on noisy classes, albeit with a lower identification AUC; secondly, as the number of clean samples increased from 30 to 300, both identification AUC and overall performance demonstrated a marked

Table 11: The repair performance on corrupted MNIST datasets with different sizes of preserved clean data. The network is LeNet.

| Size of preserved | MNIST-Label | | | | MNIST-Input | | | | MNIST-Adv | | | |
| clean data | Identification AUC | ALL | Clean | Noisy | Identification AUC | ALL | Clean | Noisy | Identification AUC | ALL | Clean | Noisy |
| --- | --- | --- | --- | --- | --- | --- | --- | --- | --- | --- | --- | --- |
| 30 | 0.76 | 83.67 | 95.99 | 69.70 | 0.68 | 96.48 | 97.07 | 95.60 | 0.57 | 96.36 | 97.07 | 95.30 |
| 100 | 0.98 | 92.82 | 93.66 | 91.55 | 0.70 | 96.24 | 96.60 | 95.70 | 0.63 | 96.46 | 96.90 | 95.80 |
| 200 | 0.99 | 95.24 | 95.83 | 94.35 | 0.70 | 96.34 | 96.47 | 96.15 | 0.68 | 96.26 | 96.50 | 95.90 |
| 300 | 0.99 | 95.60 | 95.93 | 95.10 | 0.71 | 96.16 | 96.20 | 96.10 | 0.70 | 96.44 | 96.47 | 96.40 |

improvement; and thirdly, our experiments demonstrated that 100 clean samples were sufficient for achieving effective model repair across these datasets.

# B   Omitted Technical Details in Sec. 2

# C   The Gaussian Mean Estimation Task

**Definition C.1. (Gaussian Mean Estimation)**: Given a training set $\mathcal{D}$ consisting of $n$ training data $\{x_i\}_{i=1}^{n}$ sampled from the distribution $\mathcal{U} = \mathcal{N}(\mu, \sigma^2)$ as well as $m$ training data $\{x_i\}_{i=n+1}^{n+m}$ sampled from another Gaussian distribution $\mathcal{U}' = \mathcal{N}(\mu', \sigma^2)$, where $\mu$ and $\mu'$ are the means of the two Gaussian distributions and $\sigma^2$ denotes the variance, the objective is to estimate the mean of the target distribution $\mathcal{U}$. The loss function is defined as $\mathcal{L}(x; \theta) = (\theta - x)^2$.

**Lemma C.2.** *The empirical solution of Gaussian Mean Estimation is $\hat{\theta} = \frac{1}{m+n} \sum_{i=1}^{n+m} x_i$. With probability at least $1 - p$ for some probability, the generalization error is upper bounded as*

$$|\hat{\theta} - \theta^*| \leq \alpha\sqrt{1/(n+m)} + \frac{m}{m+n}|\mu - \mu'| = B, \tag{6}$$

*where $\theta^* = \mu$ is the optimal solution and $\alpha = \sigma\sqrt{2\log(2/p)}$ is some constant.*

*Proof.* Since $\{x_i\}_{i=n+1}^{n+m}$ are sampled from $\mathcal{U}'$, $\{x_i - \mu' + \mu\}_{i=n+1}^{n+m}$ are sampled from $\mathcal{U}$. Using Gaussian tail bounds [34], we have that it holds with probability $1 - p$ that

$$\left| \frac{1}{m+n} \left( \sum_{i=1}^{n} x_i + \sum_{i=n+1}^{n+m} (x_i - \mu' + \mu) \right) - \mu \right| \leq \sigma\sqrt{\frac{2\log(2/p)}{m+n}} = \alpha\sqrt{1/(n+m)}$$

Therefore, we have

$$
\begin{aligned}
|\hat{\theta} - \theta^*| &= \left| \frac{1}{m+n} \left( \sum_{i=1}^{n} x_i + \sum_{i=n+1}^{n+m} x_i \right) - \mu \right| \\
&= \left| \frac{1}{m+n} \left( \sum_{i=1}^{n} x_i + \sum_{i=n+1}^{n+m} (x_i - \mu' + \mu) \right) - \mu - \frac{m}{n+m}(\mu - \mu') \right| \\
&\leq \left| \frac{1}{m+n} \left( \sum_{i=1}^{n} x_i + \sum_{i=n+1}^{n+m} (x_i - \mu' + \mu) \right) - \mu \right| + \frac{m}{n+m}|\mu - \mu'| \\
&\leq \alpha\sqrt{1/(n+m)} + \frac{m}{n+m}|\mu - \mu'| = B.
\end{aligned}
\tag{7}
$$

$\square$

**Remark:** Note that in cases where the value of $m$ decreases to zero, indicating the absence of any data from the detrimental distribution $\mathcal{U}'$ in the training set, the upper bound is then given by

$$B_{\text{remove}} = \alpha\sqrt{1/n}.$$

If the distribution divergence between $\mathcal{U}$ and $\mathcal{U}'$, measured by $|\mu - \mu'|$, satisfies $|\mu - \mu'| > \frac{m+n}{m}\alpha(\sqrt{1/n} - \sqrt{1/(n+m)})$, then $B_{\text{remove}} < B$. This implies that including data from $\mathcal{U}'$ in the training set negatively affects performance, and removing these data can improve the performance, given that the divergence is sufficiently large.

**Proposition C.3.** *(Recap of Proposition 2.1) In the Gaussian mean estimation task, there exists a set of alignment functions: $h(x) = x - \mu' + \tilde{\mu}$ where $|\mu - \tilde{\mu}| << |\mu - \mu'|$ such that the optimal empirical solution on the combined set of $\{x_i\}_{i=1}^n$ and $\{h(x_i)\}_{i=n+1}^{n+m}$, i.e., $\tilde{\theta} = \frac{1}{m+n}\left(\sum_{i=1}^n x_i + \sum_{i=n+1}^{n+m} h(x_i)\right)$, achieves a generalization bound $B_{align} < B_{remove}$.*

*Proof.* For any alignment function $h(x)$, similar to Eqn. (7), we have

$$
\begin{aligned}
|\tilde{\theta} - \theta^*| &= \left| \frac{1}{m+n}\left(\sum_{i=1}^n x_i + \sum_{i=n+1}^{n+m} h(x_i)\right) - \mu \right| \\
&= \left| \frac{1}{m+n}\left(\sum_{i=1}^n x_i + \sum_{i=n+1}^{n+m} (x_i - \mu' + \tilde{\mu} - \tilde{\mu} + \mu)\right) - \mu - \frac{m}{n+m}(\mu - \tilde{\mu}) \right| \\
&\leq \left| \frac{1}{m+n}\left(\sum_{i=1}^n x_i + \sum_{i=n+1}^{n+m} (x_i - \mu' + \mu)\right) - \mu \right| + \frac{m}{n+m}|\mu - \tilde{\mu}| \\
&\leq \alpha\sqrt{1/(n+m)} + \frac{m}{n+m}|\mu - \tilde{\mu}| = B_{\text{align}}.
\end{aligned}
\tag{8}
$$

If $|\mu - \tilde{\mu}| << |\mu - \mu'|$ such that $|\mu - \tilde{\mu}|$ satisfies

$$
|\mu - \tilde{\mu}| < \frac{m+n}{m}\alpha(\sqrt{1/n} - \sqrt{1/(n+m)}) < |\mu - \mu'|,
$$

we have

$$
\begin{aligned}
B_{\text{align}} &= \alpha\sqrt{1/(n+m)} + \frac{m}{n+m}|\mu - \tilde{\mu}| \\
&< \alpha\sqrt{1/(n+m)} + \frac{m}{n+m}\frac{m+n}{m}\alpha(\sqrt{1/n} - \sqrt{1/(n+m)}) \\
&= \alpha\sqrt{1/n} = B_{\text{remove}}.
\end{aligned}
\tag{9}
$$

$\square$

**Remark:** The alignment function aligns the data in $\mathcal{U}'$ with that in $\mathcal{U}$. If the alignment is effective, that is $|\mu - \tilde{\mu}| < \frac{m+n}{m}\alpha(\sqrt{1/n} - \sqrt{1/(n+m)})$, it improves performance over the results of data removal. The degree of improvement is positively correlated with the quality of alignment, as quantified by the magnitude of $|\mu - \tilde{\mu}|$.

## C.1 Cause Identification in the Gaussian Mean Estimation Task

In the Gaussian mean estimation task, we assume that all the data from the detrimental distribution $\mathcal{U}'$ are identified before model treatment. In this section, we verify the effectiveness of the metric proposed in Eqn. (1) in identifying the cause data in the training set for this task. Specifically, the metric estimates the contribution of failure by measuring the change in loss between the original trained model and an enhanced model, which is trained with a larger sample size $n$ from $\mathcal{U}$.

In the Gaussian mean estimation task, the expected loss of the data from $\mathcal{U}$ is

$$
\mathbb{E}\left[\frac{1}{n}\sum_{i=1}^{n}\mathcal{L}(x_i;\hat{\theta})\right] = \mathbb{E}\left[\mathbb{E}\frac{1}{n}\sum_{i=1}^{n}\left|x_i - \frac{1}{m+n}\sum_{j=1}^{n+m}x_j\right|^2\right]
$$

$$
= \mathbb{E}\left[\frac{1}{n}\sum_{i=1}^{n}\left|x_i - \frac{1}{m+n}\left(n\frac{1}{n}\sum_{j=1}^{n}x_j + m\frac{1}{m}\sum_{j=n+1}^{n+m}x_j\right)\right|^2\right]
$$

$$
= \mathbb{E}\left[\frac{1}{n}\sum_{i=1}^{n}\left|x_i - \frac{1}{n}\sum_{j=1}^{n}x_j + \frac{m}{m+n}\left(\frac{1}{n}\sum_{j=1}^{n}x_j - \frac{1}{m}\sum_{j=n+1}^{n+m}x_j\right)\right|^2\right]
$$

$$
= \mathbb{E}\left[\frac{1}{n}\sum_{i=1}^{n}\left|x_i - \frac{1}{n}\sum_{j=1}^{n}x_j\right|^2\right] + \mathbb{E}\left[\frac{m^2}{(m+n)^2}\left|\frac{1}{n}\sum_{j=1}^{n}x_j - \frac{1}{m}\sum_{j=n+1}^{n+m}x_j\right|^2\right]
$$

$$
= \sigma^2 + \frac{m^2}{(m+n)^2}(\mu-\mu')^2.
$$

$$(10)$$

The forth equality uses the fact that $\mathbb{E}|\xi - \mathbb{E}\xi + \varepsilon|^2 = \mathbb{E}|\xi - \mathbb{E}\xi|^2 + \varepsilon^2 - 2\mathbb{E}(\xi - \mathbb{E}\xi)\varepsilon = \mathbb{E}|\xi - \mathbb{E}\xi|^2 + \varepsilon^2$ and the fifth equality uses the fact that $\frac{1}{n}\sum_{i=1}^{n}\left|x_i - \frac{1}{n}\sum_{j=1}^{n}x_j\right|^2$, $\frac{1}{n}\sum_{j=1}^{n}x_j$ and $\frac{1}{m}\sum_{j=n+1}^{n+m}x_j$ are unbiased estimation of $\sigma^2$, $\mu$ and $\mu'$, respectively. Similarly, the expected loss of the data from $\mathcal{U}'$ is

$$
\mathbb{E}\left[\frac{1}{m}\sum_{i=n+1}^{n+m}\mathcal{L}(x_i;\hat{\theta})\right] = \sigma^2 + \frac{n^2}{(m+n)^2}(\mu-\mu')^2.
$$

If we introduce additional $\Delta n$ data to train the enhanced model $\hat{\phi}$, similar to Eqn. (10), we have

$$
\mathbb{E}\left[\frac{1}{n}\sum_{i=1}^{n}\mathcal{L}(x_i;\hat{\phi})\right] = \sigma^2 + \frac{m^2}{(m+n+\Delta n)^2}(\mu-\mu')^2,
$$

$$
\mathbb{E}\left[\frac{1}{m}\sum_{i=n+1}^{n+m}\mathcal{L}(x_i;\hat{\phi})\right] = \sigma^2 + \frac{(n+\Delta n)^2}{(m+n+\Delta n)^2}(\mu-\mu')^2.
$$

Then we have

$$
\mathbb{E}_{x_i\in\mathcal{U}}s(x_i;\hat{\theta},\hat{\phi}) = \mathbb{E}\left[\frac{1}{n}\sum_{i=1}^{n}\left(\mathcal{L}(x_i;\hat{\phi}) - \mathcal{L}(x_i;\hat{\theta})\right)\right] = \left(\frac{m^2}{(m+n+\Delta n)^2} - \frac{m^2}{(m+n)^2}\right)(\mu-\mu')^2 < 0.
$$

$$
\mathbb{E}_{x_i\in\mathcal{U}'}s(x_i;\hat{\theta},\hat{\phi}) = \mathbb{E}\left[\frac{1}{m}\sum_{i=n+1}^{n+m}\left(\mathcal{L}(x_i;\hat{\phi}) - \mathcal{L}(x_i;\hat{\theta})\right)\right] = \left(\frac{(n+\Delta n)^2}{(m+n+\Delta n)^2} - \frac{n^2}{(m+n)^2}\right)(\mu-\mu')^2 > 0.
$$

Therefore, the detrimental data can be identified by the positive value of $s(x_i;\hat{\theta},\hat{\phi})$.

Furthermore, this metric can be utilized to identify harmful data, even if they are not part of the training dataset. In the Gaussian mean estimation example, similar to Eqn. (7), we can obtain the generalization error bound to the data in $\mathcal{U}'$ as

$$
|\tilde{\theta} - u'| \leq \alpha\sqrt{1/(n+m)} + \frac{n}{n+m}|\mu - \mu'| \tag{11}
$$

As the number of training samples from $\mathcal{U}$ increases, both terms in the generalization error bound for $\mathcal{U}$ (Eqn. (7)) decrease. In contrast, for $\mathcal{U}'$, the first term $\alpha\sqrt{1/(n+m)}$ decreases, while the second

term $\frac{n}{n+m}|\mu - \mu'|$ increases. Consequently, the generalization error bound for $\mathcal{U}$ decreases more significantly than that for $\mathcal{U}'$. Given that the loss is related to the generalization bound, an increase in the number of training samples from $\mathcal{U}$ leads to a rise in the loss of most samples from $\mathcal{U}'$, while causing a decline in the loss of most samples from $\mathcal{U}$. And we can utilize the metric in Eqn. (1) to identify any detrimental data.

