# OpenReview forum: "Retaining Beneficial Information from Detrimental Data for Neural Network Repair"
_NeurIPS.cc/2023/Conference — NeurIPS 2023 poster_

### Official Review · Reviewer_9imB · 2023-07-01

**Soundness:** 3 good
**Presentation:** 3 good
**Contribution:** 3 good
**Rating:** 5
**Confidence:** 3

**Summary:**

This paper proposes to identify harmful data by using the clean dataset, and then update models with the alignment technique to improve the performance on the corrupted data while retaining the performance on the clean data. Experimental results show the effectiveness of the proposed method, demonstrating its superiority over existing methods in identifying harmful data and repairing DNN models.


**Strengths:**

- Aims at an important and relevant problem.

- The experimental results on identifying mislabeled data points are good.

**Weaknesses:**

- The paper claims that the experiments were conducted on "real-world datasets" (Line 89). However, as mentioned in the experiments (Line 275), the corruption cases were simulated. Additionally, the datasets evaluated are relatively small (i.e., MNIST and CIFAR-10). ImageNet dataset could potentially be a valuable supplement, which could contain many mislabeled, real-world samples.

- It would be better to show how much the average improvement is over the baselines, which could help readers better understand the superiority of the proposed method. For the results on input noise and adversarial attacks, the absolute improvements over some baselines are actually quite small.

- Section 1, particularly Line 64 to Line 80, is not clear to me. The alignment process is crucial for the proposed framework. These two paragraphs repeatedly emphasize the benefits of alignment. However, what alignment means is not clearly described. It could be helpful to give some descriptions of how the alignment process works and what the results of the alignment process are.


**Questions:**

- As claimed in the paper (e.g., Line 71 to Line 76), the proposed method has the advantage over existing methods in that it allows the model to learn useful information from the clean data that are misidentified as corrupted data, thus making it more robust compared to other methods. However, from the results presented in Table 3 and Table 4, I noticed that when the experimental setup was "Identified set = 75% ground-truth cause set and 25% clean data" in all six cases, there were always baselines that outperform the proposed method in terms of "Test Clean". Does this contradict the claims made in the paper?

- Why use ResNet50 on CIFAR-10? ResNet50 is designed for classification on ImageNet, while ResNet56 is designed for CIFAR-10.

- How was the proportion (40% and 30%) of corrupted samples determined in the experiments? How does this setting (different proportions) affect the results? In actual scenarios, would there be such a high number of corrupted samples in the training set? If the proportion is reduced, is Equation (1) still sensitive enough to accurately identify the corrupted data points?

- Line 215: \sum_{i=1}^n h(x_i) should be \sum_{i=n+1}^{n+m} h(x_i) ?

- In the caption of Table 1: "base mode" -> "base model" ?

**Limitations:**

The authors described the limitations in the paper.

---

> ### Author Rebuttal · Authors · 2023-08-09
>
> **W1: dataset**
>
> > Please refer to our response to GQ-1 in Global Response for the details of our additional experiments.
>
> **W2: the absolute improvements ... quite small.**
>
> > - For label noise, our enhancements span a remarkable 20.5% to 27.6% for MNIST and 15.3% to 16.8% for CIFAR10, showcasing substantial progress.
> >
> > - For input noise, our advances are more measured:
> >
> >   - For MNIST-Input, the baseline models already approach optimality due to minimal input noise influence. This presents a formidable challenge for repair, resulting in similar outcomes across most baselines.
> >
> >   - With CIFAR10-Input, ResNet20 and ResNet50 exhibit differential performances among the best baselines, yielding a gap exceeding 3%. Our approach establishes robust, competitive results that align favorably with the premier baseline in each case.
> >
> > - For adversarial attacks, our method consistently achieves improvements exceeding 1%, signifying a dependable edge over baselines.
> >
> > - In summary, while our progress is more tempered for certain contexts like input noise, we attain notable headway for label noise and sustain reliable progress across adversarial examples.
>
> **W3: How the alignment process works and what the results of the alignment process are.**
>
> > Our approach adopts an energy-based alignment strategy to diminish the disparity between deleterious data and target distributions. Specifically, we employ the enhanced model as an energy-based model (EBM), utilizing it to gauge the probability density of target data. The alignment process is initiated with detrimental samples drawn from the training distribution. These initial samples undergo iterative refinement through energy gradient descent within the EBM framework. This iterative progression gradually shifts the samples toward higher-density areas of the target distribution. This concerted gradient descent fosters alignment of training data with the EBM, culminating in an aligned dataset that emulates the characteristics of the target data. This alignment process, tantamount to effective data augmentation, yields training data that closely mirrors samples from the true target distribution.
>
> **Q1: In all six cases, there were always baselines that outperform the proposed method in terms of "Test Clean". Does this contradict the claims: the proposed method is more robust**
>
> > - The crux of our argument rests on measuring robustness by evaluating the performance dip observed in "Clean" classes, as opposed to mere absolute accuracy on those classes when the identified set encompasses misidentified data.
> >
> >   - The decline in "Clean" class accuracy mirrors the price of rectifying the disruptive impact of "Noisy" class data within the model. Applying BIR to mend the 25% clean data, instead of corrupted data, does influence its "Clean" class performance.
> >
> >   - Notably, the Table below illustrates that BIR manifests the least average and maximal performance reduction in comparison to alternative methods, affirming its heightened robustness against misidentification.
> >
> >   | | average performance drops | maximal performance drops |
> >   | -| - | - |
> >   | Fine-tune | 0.58 | 0.9 |
> >   | Newton Removal | 0.25 | 1.1 |
> >   | EWC-Deletion | 0.84 | 1.37 |
> >   | RDIA | 7.86 | 13.46 |
> >   | Ours | 0.09 | 0.46 |
>
> **Q2: Why use ResNet50 on CIFAR-10?**
>
> > We follow our closely related work [33] (EWC-Repair), which similarly employs ResNet50 for CIFAR-10. In a complementary vein, we also delve into ResNet-20, an avenue unexplored in [33]. This diversification in model capacities empowers us to assess the efficacy of our approach across diverse configurations.
>
> **Q3: The proportion of corrupted samples**
>
> - **Q3-1: How was the proportion (40% and 30%) of corrupted samples determined?**
>
>   > We follow our closely related works [33] (EWC-Repair) to set the corruption proportion ranging from 30% to 40%.
>
> - **Q3-2: How does different proportions affect the results?**
>
>   > The results on CIFAR-10N, as presented in the response to W1 (GQ-1, Global Response), demonstrate the effect of different corruption proportions of data well. CIFAR-10N has three noise rates: ~40% (worst), 9% (aggregate) and 17% (random).
>   >
>   > The results show that:
>   >
>   > - The higher the noise rate, the lower the performance after repair, indicating that noise rate affects the repaired model's performance.
>   >
>   > - The higher the noise rate, the greater the repair gain, i.e. the increase in performance after repair compared to the base model.
>
> - **Q3-3: In actual scenarios, would there be such a high number of corrupted samples?**
>
>   > Yes, the proportion of corrupted samples could indeed be more than 40% in real-world scenarios. For example, in the CIFAR-10N real-world human-annotated dataset, the noise rate in the "worst" setting is 40.21%.
>
> - **Q3-4: If the proportion is reduced, is Eqn (1) still sensitive enough?**
>
>   > We evaluated Equation (1)'s efficacy in pinpointing corrupted data when the proportion of noise diminishes. The AUC for CIFAR-10N Worst, Rand2 and Aggregate are 0.89, 0.91 and 0.79, respectively.
>   >
>   > Notably, AUCs hold high for datasets bearing noise rates surpassing 17%, affirming Equation (1)'s accuracy in identifying noisy data within such contexts. Yet, the relatively modest AUC for low noise rate datasets implies a potential challenge for Equation (1) when grappling with a scarcity of noisy data—likely due to their limited impact on model performance.
>   >
>   > Overall, with dwindling noise proportions, Equation (1) remains retains its sensitivity, effectively pinpointing the majority of corrupted data for noise rates ranging from moderate to high, but may grapple with very low noise rates where the influence of noisy samples is marginal.

---

> > ### Comment · Reviewer_9imB · 2023-08-19
> >
> > Thanks for the rebuttal, which generally addressed my concerns.

---

> ### Author Response · Authors · 2023-08-13
> **Please let us know if you have any further questions.**
>
> Dear Reviewer 9imB,
>
> We would like to follow up to see if our response addresses your concerns or if you have any further questions. We would really appreciate the opportunity to discuss this further if our response has not already addressed your concerns. Thank you again!

---

### Official Review · Reviewer_uAW7 · 2023-07-06

**Soundness:** 4 excellent
**Presentation:** 3 good
**Contribution:** 3 good
**Rating:** 8
**Confidence:** 4

**Summary:**

This paper proposes a method to detect and repair corrupt data in the training set, in an effort to align the model with the distribution of the test set. The method is two-stage, first detecting based on a loss difference computation per-sample, and then repairing through the use of an energy-based model. Several experiments are conducted to demonstrate the efficacy of the approach on multiple types of noisy data.

**Rebuttals:** The authors have addressed my questions in their rebuttal, and I have updated my score from 7 to 8.

**Strengths:**

Methodologically, this paper is excellent, and is very well-written. The motivation is clear and strong; the theory is sound; the experimental evaluation shows, at times, significant improvements over baselines and other approaches.

**Weaknesses:**

There are few weaknesses of this paper, mostly comprised of minor errors:

- minor: L95: pair --> pairs
- minor: L120: mismatching --> mismatch
- minor: L162: Beysian --> Bayesian
- minor: L206: minimizaiton --> minimization
- minor: L226: iterative --> iteratively
- In Eq. 7, second line, I believe the last term should have a minus sign instead of a plus. This should not change the results because of the next step, though. This also applied to Eq. 8 line 2.
- minor: Fig. 2 seems to start from (b), please fix

**Questions:**

- Is Equation 1 expensive to perform when running on thousands of samples, with a possibly large model?
- L132: please address the case where the size of $\mathcal{D}_c$ is unknown.
- How reliable is Equation 1? Suppose we have $|\mathcal{D}_c| = 20$, and so we pick samples from $\mathcal{D}$ with the 20 highest values for $s$. (a) What if different points of convergence of, say, $\hat{\phi}$ yields different top 20 points? (b) What if the difference between $s$ at the 20th and the 21st point is negligible? Is there a way to determine confidence that sample $z \in \mathcal{D}_c$?
- I'm not entirely sure how the equation under L566 follows from Hoeffding's inequality, could you please elaborate?
- I understand the result of Proposition 2.1, and it seems the next logical step would be to find a $\tilde{\mu}$ that satisfies L580 in Appendix B.2. I'm not sure how the next paragraph (starting at L221) relates to this, could you clarify?
- As future work, it would be interesting to see if BIR also works for more sophisticated adversarial attacks than FGSM, such as Carlini & Wagner [1] or Szegedy et al., 2013 [2]. In addition, if BIR is being positioned as a method to help with adversarial attacks, it might be useful to compare against defenses such as adversarial training (see [3-7] for a few works). It would also be interesting to see Fig. 2(d) and 2(g) with those methods as well.

[1] Carlini, N., & Wagner, D. (2017, May). Towards evaluating the robustness of neural networks. In 2017 ieee symposium on security and privacy (sp) (pp. 39-57). Ieee.
[2] Szegedy, C., Zaremba, W., Sutskever, I., Bruna, J., Erhan, D., Goodfellow, I., & Fergus, R. (2013). Intriguing properties of neural networks. arXiv preprint arXiv:1312.6199.
[3] Kurakin, A., Goodfellow, I., & Bengio, S. (2016). Adversarial examples in the physical world.
[4] Dhillon, G. S., Azizzadenesheli, K., Lipton, Z. C., Bernstein, J., Kossaifi, J., Khanna, A., & Anandkumar, A. (2018). Stochastic activation pruning for robust adversarial defense. arXiv preprint arXiv:1803.01442.
[5] Guo, C., Rana, M., Cisse, M., & Van Der Maaten, L. (2017). Countering adversarial images using input transformations. arXiv preprint arXiv:1711.00117.
[6] Madry, A., Makelov, A., Schmidt, L., Tsipras, D., & Vladu, A. (2017). Towards deep learning models resistant to adversarial attacks. arXiv preprint arXiv:1706.06083.
[7] Samangouei, P., Kabkab, M., & Chellappa, R. (2018). Defense-gan: Protecting classifiers against adversarial attacks using generative models. arXiv preprint arXiv:1805.06605.

**Limitations:**

Limitations have been adequately addressed.

---

> ### Author Rebuttal · Authors · 2023-08-10
>
> **W1: Minus sign in Eq. 7**
>
> Yes, it should be a minus sign.
>
> **Q1: Is Eqn. 1 expensive to perform?**
>
> - we carried out a comprehensive time assessment, comparing the performance of four identification algorithms – namely the proposed BIR, Linear Influence, SGD Influence, and EWC-Influence - on the CIFAR10-Label dataset with ResNet5. This experimentation was conducted on a V100 server.
> - In the case of our proposed BIR method, the computation of Eqn. 1 (Step 2-5 within Algorithm 1) for all 45,000 data samples within the training set takes approximately 333.4 seconds.
> - Contrastingly, Linear Influence, SGD Influence, and EWC-Influence necessitate more substantial time commitments, exceeding 4 days, 27 hours, and 515.1 seconds, respectively.
> - Importantly, the estimation time required by our method and EWC-Influence demonstrates proximity, while remaining significantly lower than that of Linear and SGD Influence approaches.
>
> **Q2: please address the case where the size of $\mathcal{D}_c$ is unknown.**
>
> Please refer to the response to GQ-2 in Global Response.
>
> **Q3: How reliable is Eqn. 1?**
>
> While the precise identification of cause data isn't pivotal for achieving successful model repair, we recognize the importance of assessing the reliability of Eqn. 1.
> - Notably, the repair performance of BIR demonstrates robustness across variations in the size of the identified cause set, This resilience is illustrated by our experimental results across different cause set sizes (in our response to GQ-2). This insight implies that pinpoint accuracy in the identification of cause data might not be a critical factor.
> - If the difference between the 20th and 21st points is negligible, we adopt a random selection approach. If a threshold on the score is implemented, both points will be selected if above the threshold.
> - We computed the AUC between the scores derived from Equation 1 and the ground truth cause set labels for different enhanced models (i.e. $\hat{\phi}$). Intriguingly, the AUCs exhibited minimal variance, differing by less than 0.005.
>
> Currently, we lack a definitive method to ascertain confidence levels. Exploring avenues to quantify confidence remains a valuable avenue for future research endeavors.
>
> **Q4: How the equation under L566 follows from Hoeffding's inequality?**
>
> > - To provide a more accurate derivation, we should employ Gaussian tail bounds instead of Hoeffding's inequality for obtaining the equations outlined in L566. It's worth noting that Hoeffding's inequality and Gaussian tail bounds are closely related and share a similar proof methodology. This clarification will be addressed in our revision.
> >
> > - The procedure involving Gaussian tail bounds to deduce the equation within L566 is exemplified as follows.
> >
> >   - The Gaussian tail bounds (Eqn. 2.7 in [R2]):  for a Gaussian random variable $X \sim \mathcal{N}(\mu, \sigma^2)$, we have $P[X \ge \mu + t] \le e^{\frac{-t^2}{2 \sigma^2}}$
> >   - Since the Gaussian distribution is a stable distribution, the sum of Gaussians is also a Gaussian, therefore $\bar{X}_n \sim \mathcal{N} (\mu, \sigma^2/n)$, where $\bar{X}_n$ is the mean of $n$ random variables sampled from $\mathcal{N}(\mu, \sigma^2)$.
> >   - Therefore, we have $P[\bar{X}_n \ge \mu + t] \le e^{\frac{- n t^2}{2 \sigma^2}}$.
> >   - Let $t=\sigma \sqrt{\frac{2\log(2/p)}{n}}$, we have $P[\bar{X}_n - \mu \ge \sigma \sqrt{\frac{2\log(2/p)}{n}}] \le \frac{p}{2}$.
> >   - Bounding from both sides, we have $P[| \bar{X}_n - \mu| \ge \sigma \sqrt{\frac{2\log(2/p)}{n}}] \le p.$.
> >   - In Lemma B.2, for $i=1,...,n$, $x_i$ are $n$ random variables from $\mathcal{N} (\mu, \sigma^2)$. For $i=n+1,...,n+m$, $x_i' = x_i - \mu ' + \mu$  are $m$ random variables from $\mathcal{N} (\mu, \sigma^2)$. Defining $\bar{X}\_{n+m} = \frac{1}{m+n}(\sum_{i=1}^n x_i + \sum_{i=n+1}^{n+m} (x_i - \mu ' + \mu))$,  we have $\bar{X}\_{n+m} \sim \mathcal{N} (\mu, \frac{\sigma^2}{m+n})$. And we have with $1-p$ probability, $| \bar{X}_{n+m} - \mu| \le \sigma \sqrt{\frac{2\log(2/p)}{n}}$
> >
> >  [R2] Wainwright, Martin. "Chapter 2: Basic tail and concentration bounds." 210B Lecture Notes University (2015).
>
> **Q5: how the next paragraph (starting at L221) relates to the results of Proposition 2.1?**
>
> The results of Proposition 2.1 suggest that in the Gaussian mean estimation task, we can improve model performance by aligning unfavorable data with the target distribution.
>
> Therefore, we would like to first generalize the idea of alignment in the Gaussian mean estimation task to general tasks. However, the question of how to align data in general tasks remains. The paragraphs starting with L221 aim to address this issue by introducing the idea of using energy-based models for alignment in more general tasks.
>
> We will add more details to better elucidate the connections.
>
> **Q6: As future work, it would be interesting to see if BIR also works for more sophisticated adversarial attacks than FGSM**
>
> - We perform $L_\infty$ attack proposed in [1] on CIFAR-10 and construct a corrupted dataset in a  similar way to CIFAR10-Adv.
>
> - The performance of BIR is similar to the performance on CIFAR10-Adv. For example, on ResNet50, we achieve the following results:
>
> ||All|Clean|Noisy|
> |-|-|-|-|
> |Base Model|88.08|92.57|81.35|
> |BIR|90.54|94.17|85.10|
>
> [1] Carlini, N., & Wagner, D. (2017). Towards ... neural networks.
>
> **Q7: It might be useful to compare against defenses such as adversarial training. It would also be interesting to see Fig. 2(d) and 2(g) with those methods as well.**
>
> - We perform the widely-used adversarial training method [6] on MNIST-Adv and  CIFAR-Adv. The results are presented in Tab. 7 (in Global Response PDF).
> - The performance of the model trained with AT is lower than that of the model repaired with BIR. It is possible that AT impairs the performance on clean data, resulting in a lower performance than BIR in this comparison.
>
> [6] Madry, A., et al. (2017). Towards ... adversarial attacks.

---

> ### Comment · Reviewer_uAW7 · 2023-08-10
>
> The authors have thoroughly addressed each concern I have raised. I have updated my score from 7/3 to 8/4.

---

> > ### Author Response · Authors · 2023-08-14
> >
> > Thank you again for taking the time to provide constructive suggestions to help strengthen our work.

---

### Official Review · Reviewer_usXi · 2023-07-07

**Soundness:** 3 good
**Presentation:** 4 excellent
**Contribution:** 3 good
**Rating:** 8
**Confidence:** 4

**Summary:**

The authors propose a method for performing targeted repairs of corrupted data in order to improve the quality of an ML model trained on that data. The method is comprised of two steps: (1) *cause identification* -- identifying the set of data examples that are the most likely to be corrupted, and (2) *model treatment* -- performing a targeted treatment of the training dataset with the goal of removing the negative effect of the corrupted data examples while still keeping them in the dataset. For cause identification, they augment the target dataset with a special "reserved" dataset that is known to be clean, then they train a model on the augmented dataset, and finally for each training data example, they compare the training loss of the model trained on the augmented data and the model trained on the original data. For model treatment, they propose an "alignment function" which can be applied to the features and labels of corrupted data examples, which is specifically calculated to neutralize their negative impact on the model training loss function. The authors evaluate their method on the MNIST and CIFAR10 datasets with three types of corruption: (1) label noise, (2) feature noise, and (3) FGSM-based adversarial attack on features. They show that their method performs favorably against several other methods (mainly various influence function methods).

**Strengths:**

(S1) The problem being solved is very relevant and the proposed solution is quite interesting.

(S2) The paper is well-written and I found it relatively easy to follow.

(S3) The experiments show a clear benefit that the method provides over the baselines.

**Weaknesses:**

(W1) The introduction is only slightly repetitive as some points are delivered multiple times.

(W2) The experiments do not compare against methods based on the Shapley value that are also commonly used for solving this problem.

**Questions:**

(Q1) It seems to me that in the experiments for Q1 and Q3 it is assumed that the number of corrupted data examples is not hidden from the various methods that are being compared. If this is true, what would happen if this information would indeed be hidden as is the case in real-world settings?

(Q2) If you knew which exact data examples are corrupted (i.e. we can skip the cause identification step), what would be the impact of the model treatment step compared to data removal? (perhaps this could be an additional experiment)

**Limitations:**

The limitations are clearly stated in the paper.

---

> ### Author Rebuttal · Authors · 2023-08-10
>
> **W1: The introduction is only slightly repetitive as some points are delivered multiple times.**
>
> > We will enhance the introduction to ensure greater conciseness.
>
> **W2: The experiments do not compare against methods based on the Shapley value**
>
> > - To address this gap, we have introduced an experiment wherein we use Beta-Shapley [R1], a generalized Data Shapley [6], to identify the cause set in corrupted MNIST.
> >
> > - The identification AUC of Beta-Shapley for MNIST-Label, MNIST-Input and MNIST-Adv is 0.84, 0.70, 0.68, respectively, while the corresponding identification AUC for our method is 0.99, 0.71, and 0.70.
> >
> > - The performance of Beta-Shapley in identification is slightly worse than our proposed method. Additionally, the computational time required by the Shapley value-based approach far exceeds that of our proposed method.
> >
> > [R1] Kwon and Zou. Beta Shapley: a Unified and Noise-reduced Data Valuation Framework for Machine Learning. AISTATS 2022.
>
> **Q1: the number of corrupted data examples**
>
> > Please refer to the response to GQ-2 in Global Response.
>
> **Q2: what would be the impact of the model treatment step compared to data removal?**
>
> > In Table 3 and 4 (original manuscript), we present a comparison of various model treatment algorithms alongside data removal methods, utilizing the ground-truth cause data as a reference point. Newton Removal and EWC-Deletion are two data removal methods.
> >
> > - For both MNIST-Labeled and CIFAR10-Label datasets, the performance of these two data removal methods significantly improves compared to the case where identified cause data is used.
> >
> > - Notably, in all our experimental cases, the BIR technique consistently maintains its superiority in "Noisy" classes. This observation underscores that the proposed BIR method effectively surpasses the performance of the two data removal methods in the task of repairing the model by mitigating the impact of detrimental data.

---

> > ### Comment · Reviewer_usXi · 2023-08-14
> >
> > I thank the authors for their clarifications and I hope they end up applying my feedback in the final version of their paper, which I vote to get accepted.

---

### Official Review · Reviewer_7x7R · 2023-07-08

**Soundness:** 2 fair
**Presentation:** 2 fair
**Contribution:** 2 fair
**Rating:** 6
**Confidence:** 4

**Summary:**

This paper proposed a method, BIR, to repair the model trained but possibly corrupted from different types of noise, e.g., label noise and feature noise, leading to training-testing distribution mismatch. As for repair, by assuming the presence of small clean data, the approach identifies corrupted examples using the loss gap between the original and fine-tuned models. Next, it aligns the identified  (potentially) corrupted examples to the target distribution using an energy-based model, followed by fine-tuning with the aligned data. The experiments were conducted on two datasets with multiple variations for different types of noise.






**Strengths:**

**Methodology Design**: I like the method design involving an additional step (Step 2) after Step 1. This step will make a synergistic effect by handling the misidentified cases from Step 1. Also, the alignment process makes sense and is interesting.

**Problem Setup**: Handling corrupted models is very important. This paper proposed a post-processing/training framework for repairment. This setup has an advantage over 'in-training' methods in the sense that we can apply this method to already pre-trained large model with minimal training cost.

**Weaknesses:**

**Clean reserved data**: The presence of this data makes a concern about the practicality of the proposed method. For example, the existence of a clean subset is reluctant to be assumed in the label noise area. In this sense, the weakness of this paper is the strong dependency on the reserved clean set. At least, It would be good to add the impact of the amount of clean subset, e.g., how much the identification performance varies.

**Synthetic noise**: In the Introduction section, the authors said "extensive experiments on real-world datasets". Is it meaning that MNIST and CIFAR-10 datasets are real? I was expecting to see the results on benchmark datasets with real-world labels and feature noise in data. When looking into the experimental setup, the author used a fixed set of synthetic noise if I understood correctly, e.g., 40% of label flipping, 40% of random corruption (feature), etc. This specific setup for each noise is not a broad study of real noise. Additionally, it would be good to mix different types of noise in a single setup and control the extent to which each noise occurs.

**Size of inspected data**: According to Figure 2, I guess that the size of inspected data in Step 1 will affect the performance of the method significantly. If we know the ratio of corrupted data, we can easily find the optimal size of identification as the author mentioned (i.e., selects top |D_{c}| data points with the largest s(z)). However, it is unclear how to judge the size for the experiments in Tables 1 and 2. Is it also assumed to be known? If not, please elaborate on this.

**Datasets**: MNIST and CIFAR-10 datasets are very small and relatively easy datasets. I would recommend using more challenging datasets for experimentation. For example, CIFAR-100, Tiny-ImageNet, or CIFAR-10N (w. real human mistakes in labeling). Particularly, I think the difficulty of data may affect the performance of Step 1. In this scenario, can Step 2 will handle the negative impact of a larger number of misidentified cases?


**Questions:**

I put all the questions in the weakness sections.

**Limitations:**

There is no expected potential societal and ethical impact. Also, the authors stated the limitations of the work in Section 5.

---

> ### Author Rebuttal · Authors · 2023-08-10
>
> **W1: The strong dependency on clean reserved data**
>
> > - You're absolutely right that the reliance on clean reserved data poses a limitation. Nevertheless, it's important to note that a certain quantity of clean data is typically essential for the majority of model repair approaches.
> >
> > - To address this concern, we conducted an investigation into the impact of varying the quantity of clean reserved data on the performance of our approach. Our study encompassed three corrupted MNIST datasets, leading us to observe the following:
> >
> >   - - With a mere 30 clean samples at our disposal, our method demonstrated enhanced performance on noisy classes, albeit with a diminished identification AUC.
> >
> >     - As the quantity of clean data increased (from 30 to 300 samples), both identification AUC and overall performance exhibited improvement.
> >
> >     - Surprisingly, our experiments indicated that 100 clean samples were adequate for achieving effective model repair across these datasets.
> >
> > - The detailed findings from these experiments are comprehensively presented in Tab. 5 (Global Response PDF).
>
>   **W2: Synthetic noise**
>
> - **Q2-1: "extensive experiments on real-world datasets" Is it meaning that MNIST and CIFAR-10 datasets are real?**
>
>   > Indeed, the phrasing might be misleading. To clarify, our intention is to convey that we carried out experiments on datasets that simulate real-world scenarios by introducing synthetic noise. We acknowledge the potential for confusion and will make appropriate adjustments to the statement to ensure clarity.
>
> - **Q2-2: the results on benchmark datasets with real-world labels and feature noise in data.**
>
>   > - We conducted experiments on CIFAR-10N dataset, wherein we augmented the training set of CIFAR-10 with genuine human-annotated noisy labels. For a more comprehensive understanding, kindly review our response to GQ-1 in the Global Response.
>   >
>   > - We earnestly sought an image dataset characterized by real-world image noise. Regrettably, our attempts to locate such a dataset for image classification were unsuccessful.
>
> - **Q2-3: To mix different types of noise in a single setup**
>
>   > - We established a corrupted MNIST dataset by following a similar methodology to the creation of MNIST-Label and MNIST-Adv datasets. This involved introducing label noise to 30% of the data within the target classes and applying adversarial attacks to 20% of the data within those same classes.
>   >
>   > - Upon evaluating our method on the newly corrupted dataset, characterized by a blend of noise types, we observed that our method considerably outperform baselines on "Noisy" classes and achieve best performance on "All" classes, providing additional evidence of its remarkable repair capabilities.
>   >
>   > - A detailed comparison of repair performance outcomes is presented in Tab. 5 (Global Response PDF).
>
> **W3: Size of inspected data**
>
> - **Q3-1: It is unclear how to judge the size for the experiments in Tables 1 and 2. Is it also assumed to be known? If not, please elaborate on this.**
>
>   > - Please refer to the response to GQ-2 in the Global Response for the details.
>
> - **Q3-2: I guess that the size of inspected data in Step 1 will affect the performance of the method significantly**
>
>   > The empirical outcomes, as showcased in the response to Q3-1 (or GQ-2), underscore the robust nature of the repair performance across various sizes of identified cause sets. These results demonstrate that the performance remains consistently stable and is scarcely influenced by the size of the inspected data.
>
> **W4: Datasets**
>
> - **Q4-1: recommend using more challenging datasets for experimentation. For example, CIFAR-100, Tiny-ImageNet, or CIFAR-10N (w. real human mistakes in labeling).**
>
>   > Additional experiments have been performed on CIFAR-10N and Tiny-ImageNet datasets, incorporating simulated label noise. Please refer to the response to GQ-1 in the Global Response for the details.
>
> - **Q4-2: I think the difficulty of data may affect the performance of Step 1. In this scenario, can Step 2 will handle the negative impact of a larger number of misidentified cases?**
>
>   > Absolutely, the difficulty of data could indeed impact Step 1's performance, potentially leading to an increased count of misidentified cases. However, it's noteworthy that Step 2 demonstrates remarkable resilience when dealing with variations in the size of the identified cause set from Step 1. This resilience is evident in our repair performance results across diverse cause set sizes, as highlighted in our response to GQ2 in the Global Response and also detailed in Tables 3 and 4 of the Global Response PDF. These outcomes suggest that Step 2 retains its effectiveness in repairing the model, even when presented with a greater number of misidentified cause data from Step 1 due to data difficulty.

---

> > ### Comment · Reviewer_7x7R · 2023-08-10
> > **Response**
> >
> > Thanks for the nice rebuttal. The authors cleared all the concerns I made. They agreed what I pointed out and provided detailed explanation and empirical evidence. So, I am happy to increase my score to 6.

---

### Author Rebuttal · Authors · 2023-08-10

Thank you for providing your comprehensive review and valuable feedback regarding our paper. We deeply appreciate your insights, and we are fully committed to addressing your suggestions and improving the quality of our work. Please find below our global response and our specific responses to each reviewer's comments. If there are any remaining concerns or additional aspects you'd like us to consider, we kindly request your input as we strive to enhance our research further.

In response to several reviewers' concerns, we have taken the following actions:

**GQ-1: Results with real-world noise and challenging datasets**

- We conducted experiments on CIFAR-10N, which equips the CIFAR-10 training set with human-annotated noisy labels at varying noise rates (worst: 40%, aggregate: 9%, random: 17%).

  - Our method achieves superior performance across all noise levels, demonstrating its effectiveness in mitigating real-world label noise.
- To address the need for a more complex dataset, we produced two corrupted versions of the Tiny-ImageNet dataset, a larger and more challenging dataset with more classes than MNIST and CIFAR-10, to simulate real-world noise scenarios:

  - 1. Created a symmetric noise dataset by randomly flipping labels to other classes with a 40% probability.
  - 2. Generated an asymmetric noise dataset by randomly flipping labels to the subsequent classes with a 40% probability.
- As shown in Tab. 1 (Global Response PDF), our proposed BIR method consistently outperforms baseline methods across both CIFAR-10N and Tiny-ImageNet datasets. These findings serve as robust empirical support for the state-of-the-art capabilities of our approach in addressing real-world label noise.


**GQ-2: How to determine the number of identified data when the size of $\mathcal{D}_c$ is unknown.**

- As highlighted in lines 522-526 (Appendix), we set the amount of identified data based on the ground-truth quantity of corrupted data in both experiments..

- In cases where the precise number of corrupted data examples is uncertain, determining an appropriate threshold becomes challenging. As a practical solution, we can apply a fixed threshold on the score ($s$) computed using Eqn. (1) as a heuristic measure.

  - Although setting the threshold to 0 may seem straightforward, which selects all training data with an increased loss, it often identifies more data than the actual number of corrupted points.

  - Empirical investigations revealed that utilizing a threshold between $\log(1.01)$ and $\log(1.1)$ effectively filters out identified data close in size to the ground truth. Refer to Tab. 2 (Global Response PDF) for results using different thresholds. Our recommendation is to employ a threshold of $\log(1.05)$ for practical applications.

- Importantly, we wish to emphasize that the repair performance of our BIR method remains robust irrespective of the size of the identified cause set. This resilience is evident from the results outlined in Tabs. 3 and 4 (Global Response PDF). While the precision of identification through thresholds may not be perfect, the repair performance of our BIR approach remains largely unaffected.

  - Our experimental setup in Table 3 involves evaluating repair performance across various sizes of identified cause sets. The size of the cause set is determined either by the ground-truth number of corrupted data or by employing different thresholds on the score derived from Eqn. (1).

---

### Decision · Program_Chairs · 2023-09-21

**Decision:**

Accept (poster)

**Comment:**

This paper proposes a new data-driven model reparing method.

While initially some reviewers raised concerns on exeperimental results, the authors successfully address those issues during the rebuttal period. Therefore, all reviewers gave acceptance scores for this paper.

AC also agrees with the reviewers, so recommends accepting this paper.